# Code2MCP: Transforming Code Repositories into MCP Services

## Abstract

The Model Context Protocol (MCP) aims to create a standard for how Large Language Models use tools. However, most current research focuses on selecting tools from an existing pool. A more fundamental, yet largely overlooked, problem is how to populate this pool by converting the vast number of existing software projects into MCP-compatible services. To bridge this gap, we introduce Code2MCP, an agent-based framework that automatically transforms a GitHub repository into a functional MCP service with minimal human intervention. Code2MCP employs a multi-agent workflow for code analysis, environment setup, tool function design, and service generation, enhanced by a self-correcting loop to ensure reliability. We demonstrate that Code2MCP successfully transforms open-source computing libraries in scientific fields such as bioinformatics, mathematics, and fluid dynamics that are not available in existing MCP servers. By providing a novel automated pathway to unlock GitHub, the world's largest code repository, for the MCP ecosystem, Code2MCP serves as a catalyst to significantly accelerate the protocol's adoption and practical application. The code is public at `https://anonymous.4open.science/r/Code2MCP-5B47`.

## 1 Introduction

The landscape of artificial intelligence is increasingly defined by autonomous agents that leverage Large Language Models (LLMs) to interact with external tools (Wang et al., 2024; Xi et al., 2024; Bubeck et al., 2023). To overcome the inherent limitations of LLMs in tasks requiring real-time information or precise computation, the paradigm of tool-augmented reasoning has become central (Huang et al., 2024; Hao et al., 2023). Seminal works have demonstrated that models can effectively learn to invoke external functions (Schick et al., 2023; Qin et al., 2023; Parisi et al., 2022).

However, this burgeoning ecosystem faces a fundamental scalability challenge: the $N \times M$ integration problem (Li et al., 2023; Liang et al., 2023; Qin et al., 2023; Anthropic, 2023). Each of the N models or agent applications often requires a bespoke connector for each of the M tools it must access. This results in a fragmented and inefficient system where development effort is duplicated and innovation is stifled by high integration costs (Qu et al., 2025; Shen, 2024). To address this, MCP is proposed as a universal standard that specifies how agents and tools should communicate, enabling an interoperable "plug-and-play" ecosystem (Anthropic, 2023).

In response to this integration challenge, the community's efforts have evolved, inadvertently revealing a deeper, more foundational bottleneck (Yue et al., 2025). The initial challenge is to establish the fundamental feasibility of tool use, where a limited set of tools proves the feasibility of the paradigm (Schick et al., 2023; Patil et al., 2023; Ding et al., 2025). To break past the inherent scarcity of these platforms, the focus shifts to the vast landscape of open-source repositories (Wang et al., 2025; Xie et al., 2023). This move, however, trades a scarcity problem for a chaos problem, exposing the wild non-standardization of real-world code. The MCP emerges as a direct answer to this chaos, promising a universal interface (Zhang et al., 2025; Anthropic, 2023). Yet, this leads to the critical gap: research is focused almost exclusively on the consumption side of MCP, using services from a presumed-to-exist pool (Gan & Sun, 2025), while the foundational "supply-side" problem of how to populate this pool from existing software is largely unaddressed.

However, while these efforts advance the consumption side of the problem, how agents can better use tools, they largely overlook a more fundamental bottleneck on the supply side. This supply bottleneck is not a theoretical concern but a stark reality preventing standards like MCP from achieving widespread adoption. For example, RAG-MCP (Gan & Sun, 2025) utilizes over 4,400 servers on `mcp.so`, but there are 268 million public GitHub repositories. The critical question of how to create a large and diverse pool of these standardized, agent-ready tools has been left unaddressed. This creates a major adoption gap, effectively locking away the largest software repository, GitHub, from this emerging ecosystem.

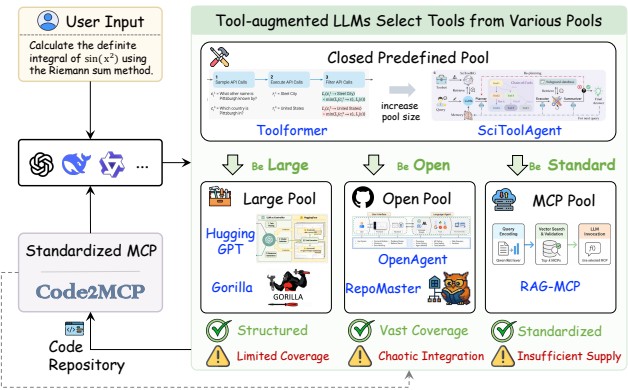

Figure 1: While most research focuses on the consumption of tools (right side), one bottleneck is their supply (left side). Code2MCP solves the supply problem by converting the code repository into a standardized MCP-compliant tool.

In this paper, we introduce Code2MCP, a new framework designed to bridge the critical tool supply gap. Code2MCP presents a blueprint for transforming any GitHub repository into a functional and documented MCP service with minimal human intervention. However, this transformation is a complex endeavor encompassing four pivotal challenges: (1) deep code comprehension to identify core functionalities, (2) reliable environment replication to ensure executability, (3) intelligent tool abstraction to design useful and valid service interfaces, and (4) robust self-correction to handle the inevitable errors throughout the process. To systematically address these challenges, as shown in Figure 2, Code2MCP implements a collaborative multi-agent system (Park et al., 2023). Unlike general-purpose coding agents (Cognition, 2024; Jimenez et al., 2024), different agents in our framework are specialized for the distinct stages of code analysis, environment setup, and API design. Crucially, the overall reliability of this workflow is ensured by an integrated Run-Review-Fix self-correction cycle, which endows the system with the ability to autonomously debug and refine the entire conversion process. The key contributions of this work are listed as follows:

- To solve the fundamental tool supply bottleneck hindering the adoption of the MCP standard, we propose Code2MCP, a novel automated framework that, to the best of our knowledge, is the first to systematically transform code repositories into agent-ready MCP services.

- The key challenge in converting code into a service is the inherent fragility of the multi-stage automation process, where an error at any step can derail the entire workflow. Thus, we introduce a novel multi-agent architecture governed by a Run-Review-Fix cycle, a self-correcting mechanism designed to systematically debug and refine the process, ensuring end-to-end reliability.

- We demonstrate the effectiveness and scalability of our framework by converting highly complex and diverse scientific libraries, covering Protein Design, Symbolic Mathematics, and Computational Fluid Dynamics, into fully functional MCP services. This provides a concrete and practical pathway to enrich the MCP ecosystem with specialized, high-value tools.

## 2 RELATED WORK

As summarized in Section 1 and Table 1, the pioneering works focus on progressively expanding the scope of tool use, from initial feasibility studies using a few predefined APIs to leveraging large, curated tool platforms, and ultimately, to the ambitious goal of directly interfacing with unstructured open-source repositories. Thus, the current bottleneck lies not in how LLMs consume tools, but in how such tools are supplied and created. In this paper, Code2MCP is designed to solve this fundamental "supply-side" problem.

**Initial explorations in LLM Tool Use.** The initial challenge is to establish the fundamental feasibility of tool use. Toolformer (Schick et al., 2023) demonstrates that an LLM can learn to invoke simple,

Table 1: A comparative summary of related works in tool-augmented LLMs.

| | Work | Core Contribution | Tool Pool | Tool Selection | MCP |
|---|---|---|---|---|---|
| **Consumer** | **Toolformer** | Teaching LLM to use external tools | 5 predefined tools | Fine-tuning | × |
| | **SciToolAgent** | Domain-specific enhancement for scientific tool utilization | KG of Scientific Tools (500+ tools) | Retrieval on KG | × |
| | **HuggingGPT** | Increase the size of tool pool | `huggingface.co` | LLM task planning | × |
| | **Gorilla** | | TorchHub, TensorHub (1600+ tools) | Retriever-aware training | × |
| | **OpenAgents** | Tool use from open-source beyond a closed pool | `github.com` | Multi-agent planning | × |
| | **RepoMaster** | | | Rule-based deep search | × |
| | **RAG-MCP** | RAG for tool selection from MCP | `MCP.so` (4,400+) | Retrieval on MCP | ✓ |
| **Supplier** | **Code2MCP** | GitHub Repo to standardized MCP | `github.com` | N/A | ✓ |

well-defined tools like a calculator via simple APIs in a self-supervised way. SciToolAgent (Ding et al., 2025) leverages knowledge graphs to orchestrate 500+ scientific tools. This proves the concept and opens a new paradigm. However, its reliance on a small, predefined set of tools is inherently unscalable and insufficient for addressing the diverse needs of real-world tasks.

**Scaling Tool Availability via Structured Platforms.** To overcome the limitation of fixed toolsets, subsequent research turns to large, curated platforms. These approaches significantly expand the number of available tools. For instance, Gorilla (Patil et al., 2023) fine-tunes models on a massive corpus of API calls from hubs like TorchHub and TensorFlow Hub. Similarly, HuggingGPT (Shen et al., 2023) positions an LLM as a controller to delegate tasks to specialized models within the Hugging Face ecosystem. While powerful, their success hinges on environments where tools are well-documented and standardized.

**Exploring Unstructured Open-Source Repositories and Challenges.** A more ambitious paradigm shift involves treating the entirety of open-source code repositories as a virtually infinite tool source. Frameworks like OpenAgents (Xie et al., 2023) and RepoMaster (Wang et al., 2025) empower agents to directly parse, reason about, and execute code within GitHub repositories. These works confront the complexity of real-world code but expose the core bottleneck: the vast majority of these repositories are not designed for programmatic use by LLM agents. They lack standardized interfaces (Zhang et al., 2024; Jin et al., 2024; Ray, 2025), forcing the agent into an ad-hoc, brittle, and unreliable process of reverse-engineering the code, setting up its environment, and managing dependencies for every single task (Zeng et al., 2024; Olausson et al., 2023). This chaotic integration process illustrates a critical failure on the tool supply side.

**The Emergence of Standardization and Unaddressed Gap.** Recognizing this chaos, the community has moved towards standardization, exemplified by the MCP. For example, RAG-MCP (Gan & Sun, 2025) explores how an agent can effectively retrieve and select the most appropriate MCP service from a pool of 4400+ available options. This approach is promising, but it presumes the existence of a rich ecosystem of MCP-compliant services (Hasan et al., 2025). This highlights a crucial, unaddressed gap: *how is this ecosystem of MCP services populated in the first place?*

## 3 METHODOLOGY: THE CODE2MCP FRAMEWORK

To achieve the goal of automatically transforming an arbitrary GitHub repository into a fully functional and reliable MCP service, we design Code2MCP, an automated framework driven by the collaboration of seven specialized agents. The entire conversion process, as depicted in Figure 2, is a multi-agent workflow that begins with code analysis, proceeds through a core Run-Review-Fix self-correction loop, and culminates in the generation of a merge-ready pull request. For completeness, a more formal description of this workflow is provided in the appendix A.3.

Suppose there exists a consumer-side work listed in Table 1 that finds a suitable GitHub repository that may solve the user's query. Code2MCP converts this repository into MCP that LLMs can call and use. This is the core difference between this "supply-side" work and the consumer-side works.

Figure 2: Overview of the Code2MCP framework. The system takes a GitHub repository URL as input and automatically generates a complete MCP service through a multi-agent workflow.

**Initialization and Analysis.** The `Download Agent` first clones the specified repository, identified by its URL $u$, into an isolated local workspace. The `Environment Agent` then replicates the runtime environment from dependency files or Dockerfiles, addressing one of the most common failure points in code conversion and supporting reliable subsequent code generation and testing.

Once the environment is ready, the `Analysis Agent` identifies tool-worthy functionalities within the codebase. It leverages the DeepWiki tool to obtain a semantic view of the code and associates code entities with their intent from documentation and comments. The output is a Code Report that summarizes candidate APIs and guides subsequent stages.

**Generation, Execution, and Self-Correction.** Given this conversion blueprint, the framework enters its core iterative loop that turns the identified functionalities into executable MCP services.

The loop starts with the `Generation Agent`, which takes the Code Report and uses an LLM to abstract the core functionalities into MCP-compliant interfaces. It creates the tool interface definitions and adapter file that connect the original code to the MCP interface, together with a basic test suite.

Once the code is generated, the `Run Agent` executes the test suite in the prepared environment to verify executability. If the tests pass, the workflow proceeds to finalization; otherwise, the `Run Agent` records the error traceback $\tau$ and forwards it to the `Review Agent`. The `Review Agent` analyzes $\tau$ together with the generated code, the Code Report, and the failing test, and diagnoses root causes such as logic errors, missing dependencies, or interface mismatches. It then formulates a correction plan $\delta$ that specifies which files and code blocks to change, and hands this plan back to the Generation Agent to re-synthesize the MCP files. This Run-Review-Fix loop repeats until tests pass or a maximum of $B$ attempts is reached.

**Finalization and Delivery.** After the core loop succeeds, the `Finalize Agent` organizes and packages the validated MCP service files. To facilitate review and adoption by the original repository maintainers, it also generates a README file explaining how to use the new MCP service. All artifacts are arranged into a reproducible directory structure under the workspace (Appendix A.2), and the agent can prepare a pull request to submit these additions back to the original repository.

## 4 EVALUATION

### 4.1 EXPERIMENTAL SETUP

**Task.** The evaluation task is: given a GitHub repository URL, Code2MCP automatically generates an MCP service for that repository, and the outcome of the conversion is judged as a success or a failure. The evaluation focuses on the overall conversion success rate across different domains and repository characteristics, and on the time and stability of Code2MCP compared with manual implementation and a GPT-4 template-based baseline on the same set of repositories.

Table 2: Per-domain summary of environment setup success, basic test success, recovery by the Run-Review-Fix (RRF) loop ("–" if none are recovered), and final MCP conversion success.

| Domain | Repos | Env succ. | Test succ. | RRF recovered | Avg rounds | MCP succ. |
|---|---|---|---|---|---|---|
| **Biomedical** | 5 | 4/5 (80%) | 2/5 (40%) | 1/2 (50%) | 1.5 | 3/5 (60%) |
| **Psychology** | 5 | 3/5 (60%) | 3/5 (60%) | 0/1 (0%) | – | 3/5 (60%) |
| **Math** | 5 | 5/5 (100%) | 3/5 (60%) | 1/2 (50%) | 2.0 | 4/5 (80%) |
| **Earth science** | 5 | 3/5 (60%) | 2/5 (40%) | 1/3 (33.3%) | 1.0 | 3/5 (60%) |
| **Chemistry** | 5 | 4/5 (80%) | 2/5 (40%) | 1/2 (50%) | 1.3 | 3/5 (60%) |
| **Physics** | 5 | 3/5 (60%) | 2/5 (40%) | 0/2 (0%) | – | 2/5 (40%) |
| **Astronomy** | 5 | 4/5 (80%) | 3/5 (60%) | 1/1 (100%) | 1.5 | 4/5 (80%) |
| **Social science** | 5 | 4/5 (80%) | 2/5 (40%) | 1/2 (50%) | 1.0 | 3/5 (60%) |
| **Linguistics** | 5 | 5/5 (100%) | 3/5 (60%) | 1/2 (50%) | 2.0 | 4/5 (80%) |
| **Econometrics** | 5 | 3/5 (60%) | 2/5 (40%) | 1/2 (50%) | 1.0 | 3/5 (60%) |
| **Overall** | 50 | 38/50 (76%) | 24/50 (48%) | 8/19 (42.1%) | 1.4 | 32/50 (64%) |

**Repositories.** Code2MCP is evaluated on 50 open-source repositories from 10 scientific and engineering domains (5 per domain), covering biomedical science, psychology, mathematical computing, earth science, chemistry, physics, astronomy, social science, linguistics, and econometrics. These repositories range from small utility libraries to large frameworks with complex dependency stacks. For each repository, the evaluation logs the environment construction result, MCP conversion result, number of generated tools, and failure types observed across all Run-Review-Fix rounds; per-repository details and the associated failure labels are summarized in a large table in the Appendix A.5, and a subset of representative scientific-computing repositories is further used for the Run-Review-Fix ablation and qualitative case studies.

**Success criterion.** A conversion is regarded as successful only if the following three conditions all hold: the runtime environment can be reconstructed according to the dependency specifications provided by the repository, the generated MCP server can be started and must pass a unified RunNode test, and at least three core tools can be invoked correctly in automatic tests and return valid outputs. If any of these conditions is not satisfied, the repository is counted as a failure.

**Failure taxonomy.** To analyze failure modes, all failed repositories are annotated with one or more labels from the following six categories:

- `env_failure`. The environment or dependencies cannot be reconstructed.
- `api_inference_error`. Systematic errors occur when inferring tool interfaces.
- `import_error`. Import paths or cross-module dependencies are misconfigured.
- `repo_internal_bug`. Bugs or version conflicts inside the original repository prevent it from running reliably even in its native environment.
- `mcp_spec_violation`. The generated JSON Schema or response structure does not conform to the MCP specification.
- `untoolable_repo`. The repository is a collection of scripts or heavily depends on interactive CLIs or GUIs, which are difficult to abstract into stable MCP tools.

**Implementation Details.** By default, Code2MCP utilizes gpt-4o-2024-05-13 as its core reasoning engine. The temperature for all models is set to 1. Code2MCP leverages `gitingest`[1] to ingest repositories into contextual prompts and fetch pre-analysis reports from `deepwiki`[2]. Case studies are conducted on servers equipped with 8 NVIDIA H100 80 GB GPUs.

## 4.2 Large-scale Repository Conversion and Failure Analysis

**Overall and per-domain success.** The overall conversion success rate is first analyzed at the domain level. In Table 2, for each domain, the number of repositories with successful environment setup

---

[1] `https://github.com/coderamp-labs/gitingest`
[2] `https://deepwiki.org`

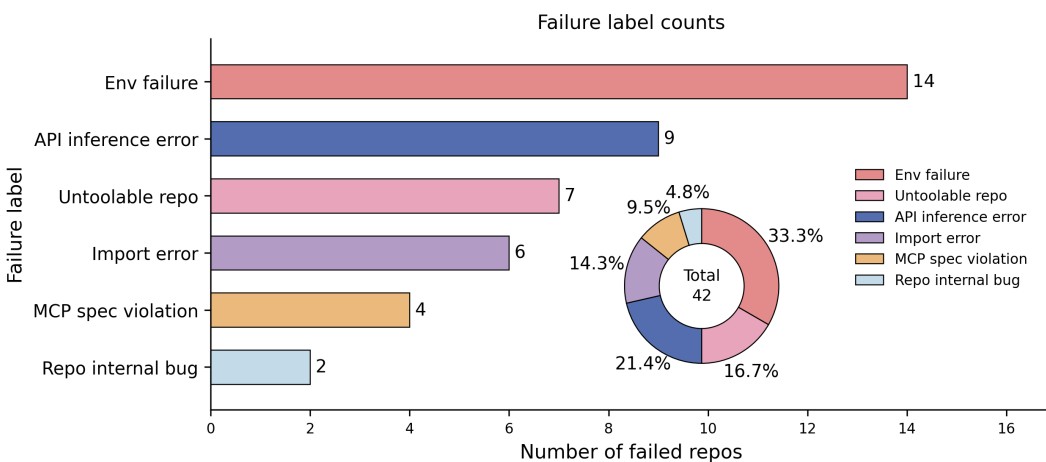

Figure 3: Distribution of 42 failure labels assigned to the 18 failed repositories. A single repository can trigger multiple failure types across different Run-Review-Fix rounds.

and basic test passing, the number of initially failing repositories recovered by the Run-Review-Fix loop and their average rounds, and the final number of successful MCP conversions. The per-domain MCP success rates range from 0.40 to 0.80: domains dominated by library-style projects, such as mathematical computing, astronomy, and linguistics, tend to have higher success rates, while domains such as physics and econometrics, which contain more complex environments and mixtures of scripts and workflow-style code, tend to have lower success rates. All statistics in the table are computed on a per-repository basis, where each repository contributes a single final outcome after the Run-Review-Fix loop. Per-repository results, including environment construction and MCP conversion outcomes as well as the number of generated tools, are summarized in the large table in the appendix and can be further grouped by domain or repository size if desired.

**Failure modes.** To understand why conversions fail on real repositories, all 18 failed repositories in the 50-repository evaluation are annotated using the six failure labels introduced in the failure taxonomy. Labels are aggregated across all Run-Review-Fix rounds, so a single repository may receive multiple labels if different error types are observed during different stages of the pipeline. The distribution of these labels is visualized in Figure 3. The labels `env_failure` and `api_inference_error` account for more than half of all failures, indicating that environment reconstruction and interface inference are the primary bottlenecks of the current workflow. The labels `import_error` and `untoolable_repo` are also relatively common, reflecting the difficulty of handling complex import paths, initialization order, and script-driven repositories in a fully automated pipeline.

**Run-Review-Fix ablation.** To assess the specific contribution of the Run-Review-Fix self-correction loop, an ablation study is conducted on a set of representative scientific-computing repositories by comparing two configurations: a single-pass configuration without the loop and the full Code2MCP pipeline with the loop enabled. Figure 4 summarizes the results. The bar chart on the left shows that the per-repository success rates increase for most repositories once the loop is enabled, often moving from medium success under a single-generation setting to success rates close to or on par with human-written wrappers. The line chart on the right plots the average number of remaining errors as a function of the number of Run-Review-Fix rounds, where "remaining errors" denote the number of failing assertions or uncaught exceptions in the unified test after each round. These results indicate that the majority of initial failures can be automatically repaired within one to three Run-Review-Fix rounds, whereas the few repositories that still fail after multiple rounds represent the main current limitation of the method.

### 4.3 COMPARISON WITH HUMAN AND GPT-4 BASELINES

The evaluation compares three configurations on the same set of representative repositories:

• a human configuration, where MCP wrappers are implemented from scratch by human developers;

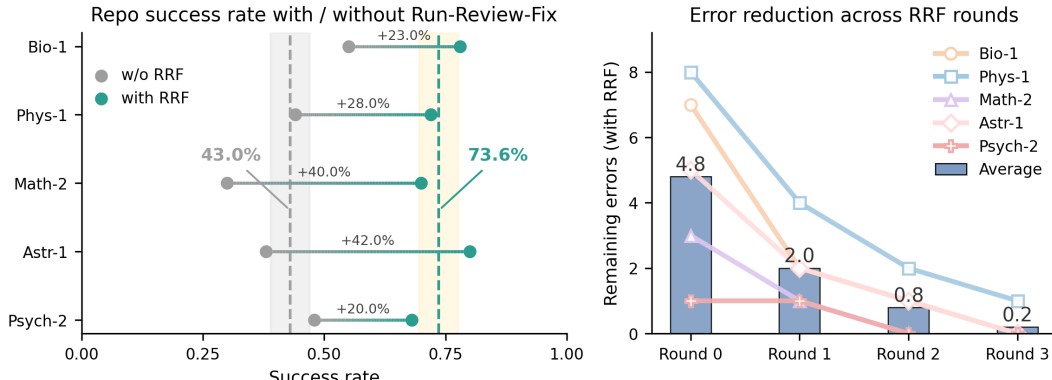

Figure 4: Repo success rates with and without Run-Review-Fix (left) and remaining errors across RRF rounds (right) for five representative scientific MCP repositories.

- a GPT-4 template configuration, where a single GPT-4 agent generates MCP wrappers;

- the Code2MCP configuration, which uses the multi-agent pipeline with the Run-Review-Fix loop.

In the human configuration, ten graduate students with at least three years of Python experience and basic familiarity with MCP concepts implement wrappers from scratch. Before the study, all participants read the MCP specification and a commented example repository. Each participant is randomly assigned several repositories, and the measured time is from the moment they start reading the repository to the point where the minimal test passes.

In the GPT-4 template configuration, a unified system prompt instructs a single GPT-4 model to generate a complete MCP service implementation based on the repository README and several key source files selected by static analysis. We use the same base model and decoding configuration as in Code2MCP, namely gpt-4o-2024-05-13 with temperature set to 1.0, to ensure a fair comparison. The model is allowed up to three dialogue turns: the first turn generates an initial implementation, and up to two additional turns may revise the code based on execution errors returned by a test runner. No additional agents or planning mechanisms are used, and the full system prompt is provided in Appendix B.1. In the Code2MCP configuration, the default multi-agent pipeline with

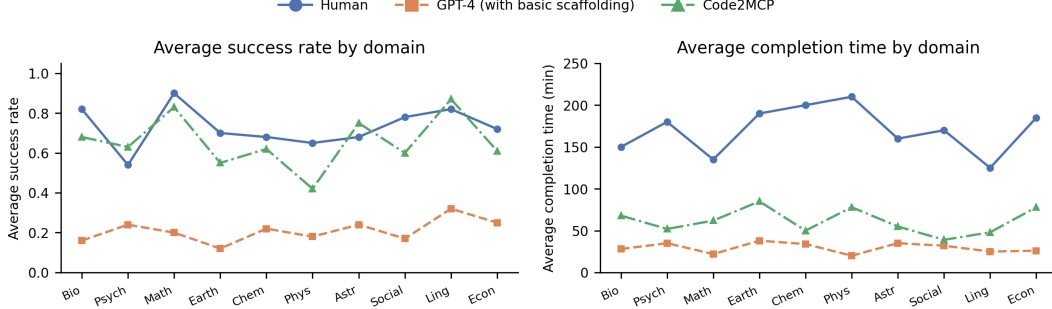

Figure 5: Average task success rate (left) and average completion time (right) across the ten scientific domains for the three configurations: Human experts, GPT-4 with basic scaffolding, and Code2MCP.

the Run-Review-Fix loop is used. For each repository, we record the final success status and the completion time, defined as the end-to-end wall-clock time starting from the beginning of environment setup and ending when the unified MCP test first passes or the run is terminated; see Appendix B.2 for details. Figure 5 summarizes the average task success rates and completion times of the three configurations across the ten scientific domains. For each configuration and domain, task success rate is defined as the fraction of successful attempts over all attempts on representative repositories in that domain. Completion time is the average end-to-end wall-clock time per attempt, measured from the beginning of environment setup until the minimal MCP test first passes or the run is terminated.

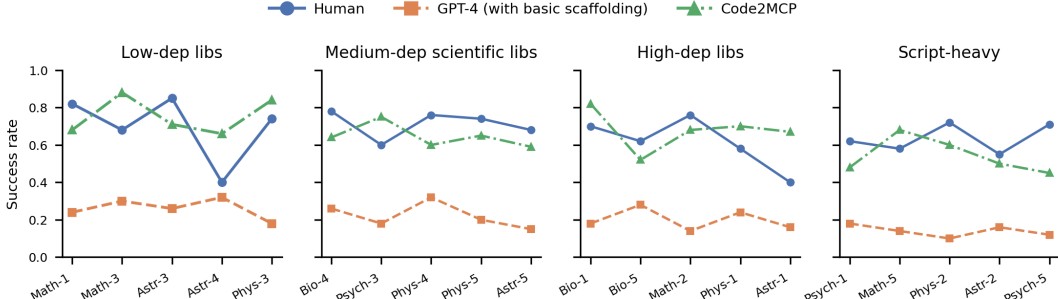

Figure 6: Task success rates of Human experts, GPT-4 with a template, and Code2MCP on representative repositories in the four repository groups. The x-axis lists representative repositories in each group, labeled by domain abbreviation and index (e.g., Math-1).

The overall trend is as follows: Code2MCP achieves task success rates close to human implementations in most domains, and consistently higher than the GPT-4 template configuration. In terms of completion time, human implementations are typically on the order of hours, while Code2MCP completes in tens of minutes; the GPT-4 template configuration is the fastest but has lower success rates.

When repositories are grouped by dependency complexity and project type into low-dependency libraries, medium-dependency scientific libraries, high-dependency libraries, and script-heavy projects, Figures 6 and 7 further break down the same comparison on representative repositories within each group. The bucketing rules are detailed in Appendix B.2.

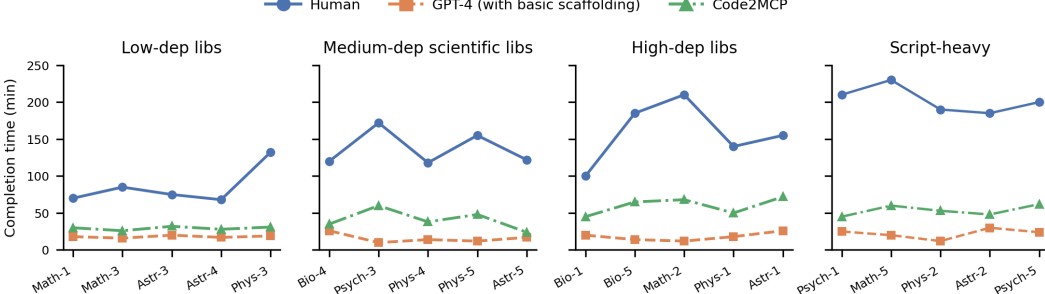

Figure 7: Completion time of Human experts, GPT-4 with a template, and Code2MCP on representative repositories in the four repository groups.

## 4.4 INTEGRATION WITH EXISTING TOOL SYSTEMS

This subsection summarizes how Code2MCP integrates with existing tool ecosystems by combining it with RepoMaster and OpenAgents on a shared set of tool-usage tasks.

**RepoMaster + Code2MCP.** In the RepoMaster setting, code-understanding and tool-usage tasks are run in two configurations: using RepoMaster alone to operate at the source-code level, and using RepoMaster together with Code2MCP, where RepoMaster first selects relevant repositories and then invokes the corresponding MCP tools. Figure 8 shows that the configuration with Code2MCP achieves higher task success rates and requires fewer interaction steps (see Appendix B.2 for metric definitions), suggesting that delegating part of the work to standardized MCP tools improves efficiency without modifying the upstream agent policy.

**OpenAgents + Code2MCP.** In the OpenAgents setting, cross-domain tasks are evaluated under two configurations: using OpenAgents alone, and using OpenAgents with MCP tools automatically generated by Code2MCP added to the tool pool. As shown in Figure 9, we report the task success rate and coverage, where coverage and success rate are defined in Appendix B.2. Adding Code2MCP

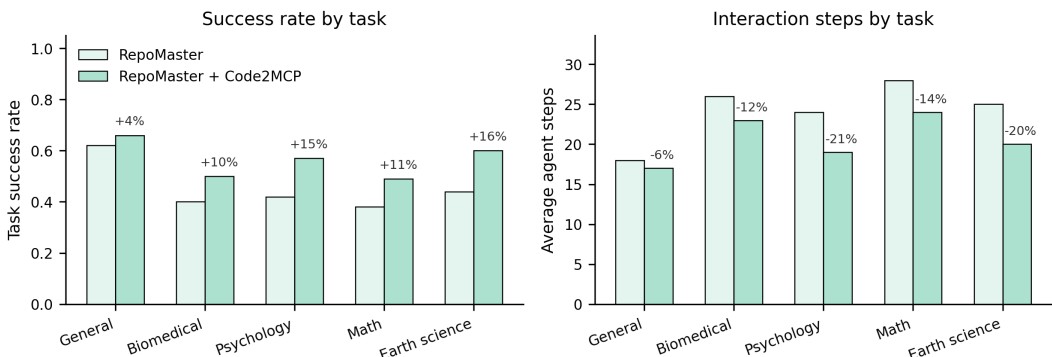

Figure 8: RepoMaster with and without Code2MCP tools: task success rate (left) and average interaction steps (right) across task groups.

tools increases both metrics across all task groups, indicating that a richer MCP tool pool can be effectively exploited by existing planning and retrieval strategies.

Overall, RepoMaster and OpenAgents handle tool discovery and code retrieval on the consumer side, while Code2MCP supplies additional MCP services on the supply side. Under the same front-end selection and planning strategies, integrating Code2MCP enriches the tool pool and yields measurable improvements in end-to-end task performance.

## 4.5 Case Studies: Protein, Math, and Computational Fluid Dynamics

This subsection highlights three representative repositories from the 50-repository evaluation set: the biomedical protein modeling library ESM, the symbolic mathematics library SymPy, and the CFD framework Foam-Agent built on OpenFOAM, illustrating the kinds of MCP tools Code2MCP generates and how agents use them in practice.

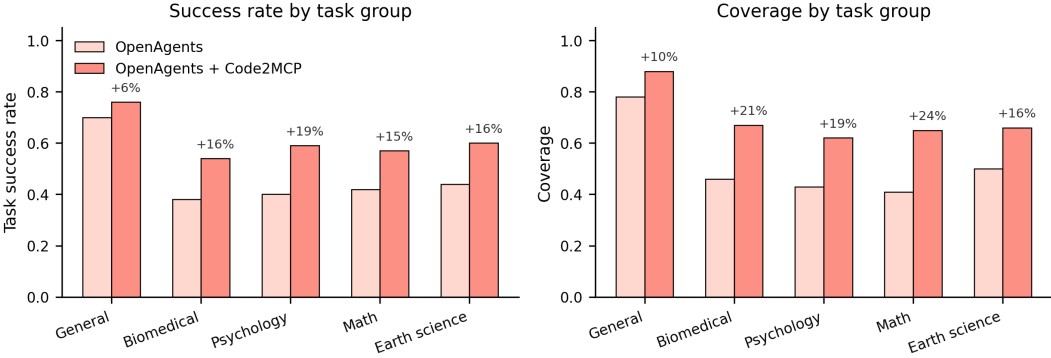

Figure 9: OpenAgents with and without Code2MCP tools: task success rate (left) and coverage (right) across general and scientific task groups.

**Bridging protein science with AI agents.** In protein science, models such as AlphaFold greatly improve structure prediction, but using them in everyday research typically requires substantial scripting and environment setup. ESM models complement AlphaFold with efficient sequence modeling and zero-shot variant effect prediction, yet traditionally still require multi-step Python pipelines for computing physicochemical properties, predicting structures, and analyzing mutations. Code2MCP converts ESM into an MCP service by exposing tools such as `Analyze_sequence`, `Predict_structure`, and `Predict_variant_effect`. We provide a qualitative example of using ESM-based MCP tools for protein analysis in Appendix A.5.

**Enhancing mathematical reliability in AI agents.** For mathematical computing, large language models are prone to errors in symbolic derivations and exact calculations, whereas libraries such as SymPy already provide reliable implementations. Code2MCP organizes common capabilities from

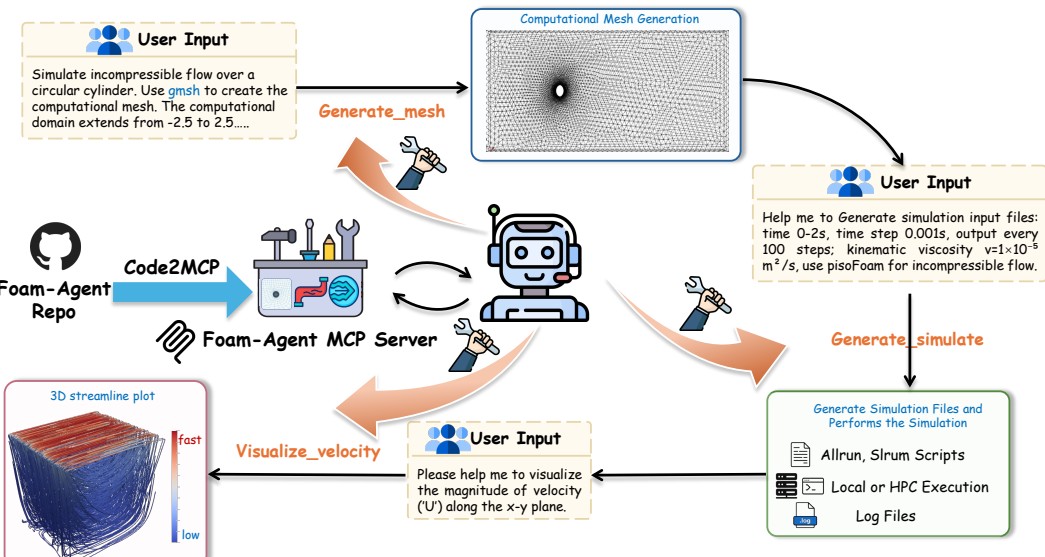

Figure 10: An AI agent orchestrates a CFD simulation pipeline by invoking `Generate_mesh`, `Generate_simulate`, and `Visualize_velocity` functions to guide a user through the entire process, from initial mesh generation and simulation setup to the final velocity visualization.

SymPy into a mathematical MCP service with tools for limits and integration, matrix operations, symbolic simplification, and transforms such as Fourier and Laplace. When asked to compute an integral or the volume of a solid of revolution, an agent simply invokes the corresponding tool and returns an exact answer, relying on SymPy as a backend instead of ad-hoc prompt-based reasoning.

**Automating CFD simulation for AI agents.** CFD workflows typically involve mesh generation, solver configuration, execution, and post-processing, traditionally coordinated by engineers through shell scripts and configuration files. Foam-Agent provides a higher-level interface to Open-FOAM, and Code2MCP further turns it into an MCP service with tools such as `Generate_mesh`, `Generate_simulate`, and `Visualize_velocity`. As illustrated in Figure 10, a user describes the target geometry and physical parameters in natural language, and the agent calls these tools in sequence to construct the mesh, run the simulation, and visualize the velocity field.

## 5 CONCLUSION

Addressing the key challenge of insufficient tool supply in the MCP ecosystem, this paper introduces Code2MCP, a framework that automatically converts GitHub repositories into functional MCP services. The framework employs a multi-agent workflow for code comprehension, environment reconstruction, and tool abstraction, augmented by a "Run-Review-Fix" self-correction loop that improves end-to-end reliability across diverse scientific libraries. Our evaluation on 50 repositories shows that Code2MCP can robustly expose high-value APIs as MCP tools while also revealing clear failure modes and boundary conditions. This work presents an automated pathway for connecting existing codebases to the agent tool ecosystem, and we leave systematically hardening security and building standardized large-scale MCP benchmarks from these converted services as important directions for future work.

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

# A   APPENDIX

## A.1   USE OF LARGE LANGUAGE MODELS

During manuscript preparation, large language models (LLMs) are used solely as general-purpose writing assistants for grammar checking, wording refinement, and improving clarity. LLMs don't contribute to research ideation, methodological design, or experimental execution. All suggestions produced by LLMs are reviewed, edited, and vetted by the authors, who take full responsibility for the entire content of the paper.

## A.2   OUTPUT DIRECTORY STRUCTURE

Figure 11 shows the final directory layout generated by Code2MCP, where the original repository and the synthesized MCP artifacts are organized under a unified workspace.

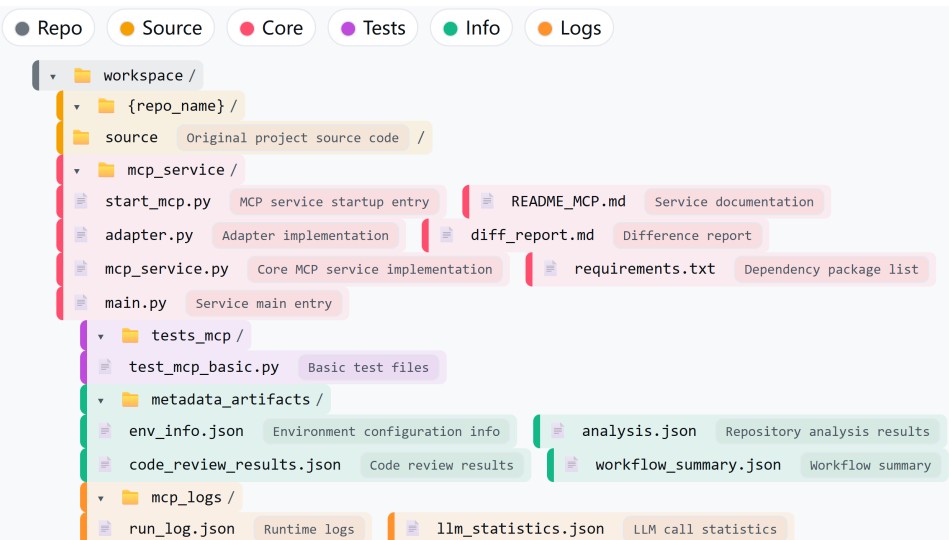

Figure 11: The complete output directory structure generated by the Code2MCP framework. The top-level `workspace` contains the original repository alongside all generated artifacts within the `mcp_output` directory.

## A.3   DETAILED CONVERSION PIPELINE AND ALGORITHM

This section outlines the roles of the specialized agents within the Code2MCP framework and their coordination as shown in Algorithm 1, where Download and Environment initialize the workspace, Analysis and Generation construct the MCP service, Run and Review form the Run-Review-Fix loop, and Finalize produces the final deliverables. The following are placeholders for their system prompts.

## A.4   AGENT ROLES AND SYSTEM PROMPTS

This section outlines the roles of the specialized agents within the Code2MCP framework. The following are placeholders for their system prompts.

**Environment Agent** This agent rapidly provisions a minimal, isolated runtime for the repository, with minimal dependencies and a short smoke test; if setup fails, propose one lightweight, auditable fallback without modifying the repository; keep all steps reproducible and pragmatic.

---

**Algorithm 1** The Code2MCP Framework

---

1: **Input:** GitHub repository URL $u$
2: `Download Agent`: Clone repository into an isolated workspace.
3: `Environment Agent`: Replicate runtime environment from dependency files.
4: `Analysis Agent`: Analyze codebase $\rightarrow$ generate detailed Code Report.
5: `Generation Agent`: Synthesize initial MCP files (`mcp_service.py`, `adapter.py`, tests) based on Code Report.
6: $r \leftarrow 0$; $success \leftarrow$ false
7: **while** $\neg success \wedge r < B$ **do**
8:    `Run Agent`: Execute test suite; collect error traceback $\tau$ on failure.
9:    **if** all tests pass **then**
10:      $success \leftarrow$ true
11:    **else**
12:      `Review Agent`: Analyze traceback $\tau$ and generate correction plan $\delta$.
13:      `Generation Agent`: Re-synthesize MCP files using the Code Report and correction plan $\delta$.
14:      $r \leftarrow r + 1$
15:    **end if**
16: **end while**
17: `Finalize Agent`: Package service files, generate README, and create a Pull Request.
18: **Output:** A merge-ready Pull Request containing the functional MCP service.

---

### Environment System Prompt

- Prefer Conda; use venv only if Conda is unavailable or clearly unsuitable.
- Detect dependency sources by priority: environment.yml > requirements.txt > pyproject.toml > setup files; never guess hidden dependencies.
- Pin versions when explicit; otherwise install the minimal viable set. Prefer CPU wheels unless GPU is explicitly required.
- Normalize cross-platform behavior; avoid absolute paths; use relative POSIX-like paths; ensure UTF-8 locale.
- Smoke test: print Python version and platform; import fastmcp; attempt to import the project's top-level package or a primary CLI; exit code 0 indicates pass.
- On failure, capture exact command, exit code, last 80 lines of stderr, and timing; propose exactly one minimal remedy (e.g., switch to venv, install a single missing package, try one version pin, extend timeout once).
- Apply at most one fallback; never change repository code; do not write outside the workspace; do not weaken security (e.g., no SSL bypass).
- Cache wheels where possible; avoid global pollution; record reproducible commands and resolved versions.
- Default to offline validation; if network is strictly required, justify briefly and bound the scope.
- Emit a compact environment report (platform, Python, manager, explicit deps, resolved pins, pass/fail).

**Code Analysis Expert** This agent performs static analysis to shape the repository into a compact, high-value capability surface, selecting stable public functionality, filtering out test/demo code, and producing a concise plan aligned with predefined domains, categories, and solvers.

### Code Analysis System Prompt

- Ingest repository signals (structure, import graph, README/docstrings, CLI entry points; DeepWiki if available) to identify stable public APIs suitable as MCP tools.
- Prefer documented, side-effect-bounded surfaces; exclude tests, internals, notebooks, long demos unless clearly valuable and controllable.
- Define crisp tool boundaries: explicit inputs/outputs, preconditions/postconditions, resource needs (CPU/GPU/memory/time), and I/O constraints.
- Note minimal adapter needs (path normalization, dtype coercion, lazy imports) and hazards (network access, file mutation, global state).

- Summarize fragilities (optional deps, platform quirks) and propose guards (timeouts, argument validation, deterministic seeds).
- Also produce a case description: case name, case domain, case category, and case solver, using a consistent taxonomy across repositories.
- Output a compact plan for generation: candidate tools (name, brief description, inputs with types/defaults, outputs, idempotency, side effects) and environment assumptions. Keep it actionable and minimal.

**Code Generation Expert** This agent synthesizes a robust MCP service from the analysis plan with clean design, consistent interfaces, graceful failure handling, and immediate executability, defining clear tool endpoints, enforcing explicit typed parameters and standardized returns, and avoiding test or example tooling.

### Code Generation System Prompt

- Produce clean, runnable Python (no Markdown fences). Use FastMCP to build the MCP service.
- Implement create_app() that returns the service; register tools with concise names and user-facing descriptions.
- For every tool: explicit, typed parameters (no *args/**kwargs); validate inputs; JSON-serializable outputs.
- Standard return shape: success: bool, result: any or null, error: string or null.
- Handle optional dependencies via lazy imports; emit helpful errors without crashing the service; prefer CPU fallbacks when reasonable.
- Ensure deterministic defaults (fixed seeds when relevant); avoid hidden global state; restrict file I/O to the workspace with existence/size/extension checks.
- Design for cross-platform paths; avoid shell-specific behavior; bound execution time and memory.
- Do not generate tests as tools. Expose a small set of high-value, composable endpoints; avoid overexposing internals.
- Add lightweight logging (tool name, argument schema, durations) and minimal version metadata to aid troubleshooting.

**Senior Software Engineer** This agent diagnoses failures and applies the smallest auditable change that restores correctness while preserving public contracts, deciding between direct fix and regeneration, using strict complete-file replacement, and avoiding multi-file edits or prose in outputs.

### Review & Auto-Fix System Prompt

- Triage failures: import/env, type/contract, path/I-O, dependency/version, timeout/perf, platform.
- Choose minimal direct fix vs. regeneration; provide a one-line rationale and confidence (low/med/high). Prefer adapter-boundary mitigations (lazy import, existence checks, parameter coercion).
- Apply strict complete-file replacement for the single target file; return pure code only; do not alter unrelated sections.
- Preserve interface contracts and standardized error shapes; add narrow guards instead of broad catches.
- Enforce cross-platform path handling and deterministic behavior; do not introduce external network calls or new side effects.
- Optionally add a tiny internal sanity check if it prevents recurrence without bloat.
- Bound attempts (<= B). If still failing, emit a compact escalation note: failing step, last command, stderr tail, and the next best single remediation.

**Final Agent** This agent consolidates artifacts and workflow logs into developer-facing documentation and delivery notes that are precise, reproducible, and integration-ready, producing a concise README with installation, quick start, key tools, troubleshooting guidance, and references.

### Final Agent System Prompt

- Write a concise developer README (Markdown) including:
1) Overview and value; roles of MCP and FastMCP; supported OS.
2) Minimal reproducible install (Conda/venv), commands, pinned dependencies, offline notes; Windows PowerShell and Linux shell variants.
3) Quick start to launch the service and call 2–3 key tools with copy-pasteable snippets and basic

error handling.
4) Tool list: endpoint name, one-line description, key parameters (types/defaults), return shape, idempotency/side effects, typical runtime class.
5) Troubleshooting: environment/import issues, optional deps, timeouts, path problems, permissions, CPU/GPU enablement; any bounded network caveats.
6) Reproducibility and telemetry: how to capture environment report, versions, minimal repro commands; where logs/artifacts live.
7) License and compliance notes: repository license, usage constraints, safety guardrails.
- Keep structure clear, steps verifiable, and assumptions minimal; prioritize essentials for successful adoption and integration.

## A.5 Additional Protein Case Study: ESM

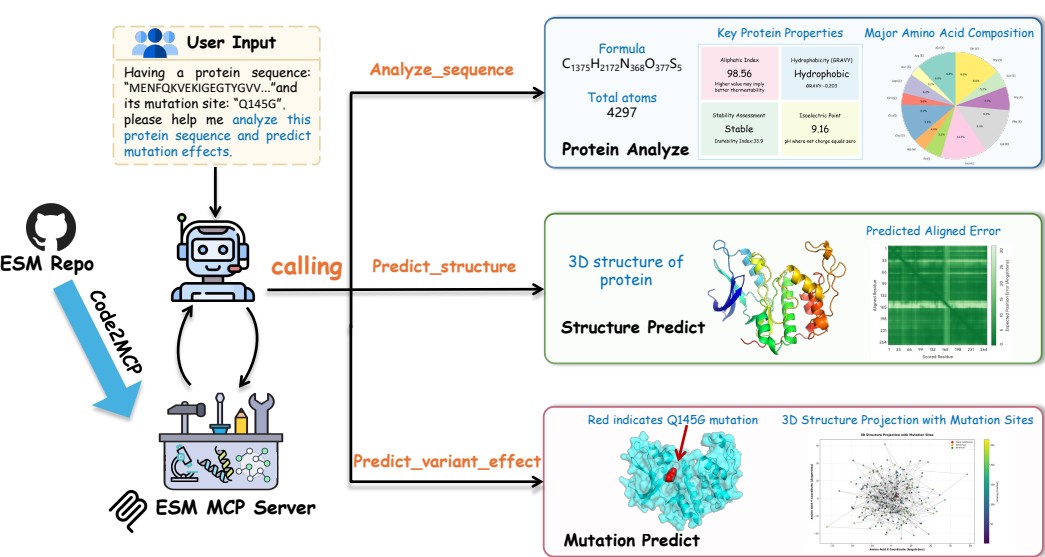

Figure 12: An AI agent processes a user's query containing a protein sequence and mutation. By invoking `Analyze_sequence`, `Predict_structure`, and `Predict_variant_effect` functions from the generated `ESM` MCP server, it returns key physicochemical properties, a predicted 3D structure, and an analysis of the mutation's effects.

## B Details of Evaluation

### B.1 GPT-4 Template Baseline Prompt

In the GPT-4 template baseline, we use a single system prompt that instructs one GPT-4 model to generate a complete MCP service implementation for a given repository. The full prompt is given below.

---

**GPT-4 Template Baseline System Prompt**

You are a single GPT-4 model acting as an autonomous MCP wrapper generator for a Python code repository. You receive: - A short task description. - The repository README (possibly truncated). - A list of key source files selected by static analysis, each with its file path and content. - A minimal description of the MCP protocol and the FastMCP library.

Your goal is to produce a single, self-contained Python MCP server that exposes a small set of high-value tools from this repository.

Requirements: 1. **Design of tools** - Identify 3–8 core, stable capabilities that are useful to expose as tools. - For each tool, choose a concise, descriptive name and a one-sentence natural-language description. - Specify explicit, typed parameters (no `*args` or `**kwargs`); include default values where reasonable. - Define clear, JSON-serializable return values. Avoid returning raw Python objects that cannot be serialized.

---

> 2. **Implementation with FastMCP** - Use the FastMCP library to implement the MCP server. - Implement a `create_app()` function that constructs and returns the MCP application. - Register each tool with FastMCP, binding it to the underlying repository functions or classes. - Prefer importing from public, documented APIs instead of private or test modules. - Use lazy imports for heavy optional dependencies when necessary, and return a helpful error message if a required package is missing.
> 3. **Robustness and safety** - Validate all inputs (types, value ranges, file existence) before calling library functions. - Handle exceptions inside each tool and return a standardized error object instead of raising uncaught exceptions. - The standardized response format for each tool is: { `"success":  bool`, `"result":  any or null`, `"error":  string or null` }. - Avoid network access unless the repository explicitly requires it. - Restrict file I/O to the workspace; do not write to absolute paths outside the project.
> 4. **Coding style** - Write clean, idiomatic Python 3 code with clear function boundaries. - Add short docstrings for each tool describing its purpose, parameters, and return value. - Do not generate tests, examples, or documentation as part of the MCP server file. - Do not include Markdown fences such as ""'; output only raw Python code.
> 5. **Output format** - Return a *single* Python file containing the complete MCP server implementation. - Do not include explanations, comments to the user, or multiple files; only the final server code that can be saved as e.g. `mcp_service.py`.

## B.2 Evaluation Metrics and Bucketing Rules

This section defines the evaluation metrics and repository bucketing rules used in the main experiments.

**Interaction steps.** In the RepoMaster experiments, an interaction step is defined as a single high-level agent action that changes the external state of the system. Concretely, we count as one step any of the following: (1) invoking a tool (including Code2MCP-generated MCP tools and built-in RepoMaster tools), (2) editing source code files, or (3) executing the target program or tests. The total number of interaction steps for a task is the length of the resulting action sequence.

**Task Set Description.** In the integration experiments with RepoMaster and OpenAgents, we use a fixed set of code-understanding and tool-usage tasks that are primarily derived from public benchmarks and representative scenarios in prior work, with a small number of additional scientific-computing cases to ensure coverage of both general and scientific settings. At this stage, this task set remains a relatively small, research-oriented setup rather than a fully standardized large-scale benchmark, and constructing a more systematic MCP task benchmark is left as future work.

**Task success rate and coverage.** In the OpenAgents experiments, a task is considered successful if the agent produces a final answer that passes automatic correctness checks. Task success rate is the fraction of tasks in a group that are successful. Coverage is the fraction of tasks in a group for which the agent produces any final answer that can be checked by the automatic evaluator, regardless of whether the answer is ultimately judged correct.

**Repository buckets.** To analyze the impact of dependency complexity and project style, we group repositories into four buckets based on static metadata:

- **Low-dependency libraries:** at most 10 Python dependencies inferred from `requirements.txt`, `environment.yml`, or `pyproject.toml`, and primarily library-style public APIs.
- **Medium-dependency scientific libraries:** between 11 and 30 Python dependencies with a library-style API and no substantial external system requirements.
- **High-dependency libraries:** more than 30 Python dependencies or additional system-level requirements (e.g., MPI stacks, GPU-accelerated frameworks, or complex external services).
- **Script-heavy projects:** projects whose primary functionality is exposed via CLI scripts, notebooks, or workflow-style pipelines rather than a stable library API; these are identified using repository structure, entry points, and README descriptions.

## C  Per-repository Results

Table 3: Repository identifiers, domains, and final outcomes.

| ID | Domain | Repo | Final_status | Main_failure_type |
|----|--------|------|--------------|-------------------|
| B1 | Biomedical | facebookresearch/esm | success | – |
| B2 | Biomedical | biocore/scikit-bio | fail | env_failure |
| B3 | Biomedical | Biopython/Biopython | fail | api_inference_error |
| B4 | Biomedical | pysam-developers/pysam | success | – |
| B5 | Biomedical | deepchem/deepchem | success | – |
| P1 | Psychology | psychopy/psychopy | fail | untoolable_repo |
| P2 | Psychology | pymc-devs/pymc | success | – |
| P3 | Psychology | sahahn/BPt | success | – |
| P4 | Psychology | mne-tools/mne-python | fail | api_inference_error |
| P5 | Psychology | neuropsychology/NeuroKit | success | – |
| M1 | Math | sympy/sympy | success | – |
| M2 | Math | scipy/scipy | fail | api_inference_error |
| M3 | Math | fredrik-johansson/mpmath | success | – |
| M4 | Math | cvxpy/cvxpy | success | – |
| M5 | Math | sagemath/sage | success | – |
| Ea1 | Earth Science | obspy/obspy | success | – |
| Ea2 | Earth Science | Unidata/MetPy | fail | mcp_spec_violation |
| Ea3 | Earth Science | pyproj4/pyproj | fail | repo_internal_bug |
| Ea4 | Earth Science | mapbox/rasterio | success | – |
| Ea5 | Earth Science | geopandas/geopandas | success | – |
| A1 | Astronomy | astropy/astropy | success | – |
| A2 | Astronomy | sunpy/sunpy | fail | repo_internal_bug |
| A3 | Astronomy | lightkurve/lightkurve | success | – |
| A4 | Astronomy | astroML/astroML | success | – |
| A5 | Astronomy | astropy/astroquery | success | – |
| C1 | Chemistry | rdkit/rdkit | success | – |
| C2 | Chemistry | openbabel/openbabel | fail | env_failure |
| C3 | Chemistry | bjodah/chempy | success | – |
| C4 | Chemistry | pyscf/pyscf | fail | api_inference_error |
| C5 | Chemistry | cclib/cclib | success | – |
| Ph1 | Physics | csml-rpi/Foam-Agent | success | – |
| Ph2 | Physics | PlasmaPy/PlasmaPy | fail | env_failure |
| Ph3 | Physics | pydy/pydy | fail | – |
| Ph4 | Physics | PyAbel/PyAbel | success | – |
| Ph5 | Physics | scikit-hep/scikit-hep | fail | untoolable_repo |
| S1 | Social Science | networkx/networkx | success | – |
| S2 | Social Science | igraph/python-igraph | success | – |
| S3 | Social Science | networkit/networkit | fail | env_failure |
| S4 | Social Science | snap-stanford/snap-python | fail | untoolable_repo |
| S5 | Social Science | datamade/dedupe | success | – |
| L1 | Linguistics | nltk/nltk | success | – |
| L2 | Linguistics | explosion/spaCy | success | – |
| L3 | Linguistics | stanfordnlp/stanza | success | – |
| L4 | Linguistics | RaRe-Technologies/gensim | fail | import_error |
| L5 | Linguistics | flairNLP/flair | success | – |
| E1 | Econometrics | statsmodels/statsmodels | success | – |
| E2 | Econometrics | alkaline-ml/pmdarima | success | – |
| E3 | Econometrics | facebook/prophet | fail | env_failure |
| E4 | Econometrics | blue-yonder/tsfresh | fail | repo_internal_bug |
| E5 | Econometrics | pydata/xarray | success | – |

Table 4: Per-repository structural statistics (size, dependencies, and tests).

| ID | total_files | total_size | Size_bucket | Dependency_complexity | Has_tests | Has_packaging |
|---|---|---|---|---|---|---|
| B1 | 471 | 33M | Medium | Medium | Yes | Yes |
| B2 | 874 | 9M | Medium | Medium | Yes | Yes |
| B3 | 1146 | 95M | Medium | High | Yes | Yes |
| B4 | 561 | 13M | Medium | Medium | Yes | Yes |
| B5 | 1411 | 136M | Medium | High | Yes | Yes |
| P1 | 3542 | 65M | Large | High | Yes | Yes |
| P2 | 371 | 17M | Medium | High | Yes | Yes |
| P3 | 2669 | 39M | Large | Medium | Yes | Yes |
| P4 | 1385 | 26M | Medium | High | No | No |
| P5 | 496 | 52M | Medium | Medium | Yes | Yes |
| M1 | 1968 | 29M | Medium | Low | Yes | Yes |
| M2 | 3027 | 81M | Large | High | Yes | Yes |
| M3 | 197 | 2M | Small | Low | No | Yes |
| M4 | 1079 | 38M | Medium | Medium | Yes | Yes |
| M5 | 5331 | 113M | Large | High | Yes | Yes |
| Ea1 | 2117 | 34M | Large | High | Yes | Yes |
| Ea2 | 595 | 99M | Medium | Medium | Yes | Yes |
| Ea3 | 144 | 1M | Small | Medium | No | Yes |
| Ea4 | 382 | 17M | Medium | Medium | Yes | Yes |
| Ea5 | 351 | 11M | Medium | Medium | Yes | Yes |
| A1 | 1879 | 24M | Large | High | Yes | Yes |
| A2 | 877 | 10M | Medium | Medium | Yes | Yes |
| A3 | 217 | 8M | Small | Medium | No | Yes |
| A4 | 192 | 1M | Small | Low | Yes | Yes |
| A5 | 960 | 23M | Medium | Medium | Yes | Yes |
| C1 | 5740 | 130M | Large | High | Yes | Yes |
| C2 | 10000 | 63M | Large | High | Yes | Yes |
| C3 | 213 | 1M | Small | Low | No | Yes |
| C4 | 2114 | 56M | Medium | High | Yes | Yes |
| C5 | 1508 | 80M | Large | Medium | Yes | Yes |
| Ph1 | 44 | 0.2M | Small | Medium | Yes | Yes |
| Ph2 | 498 | 16M | Medium | Medium | Yes | Yes |
| Ph3 | 263 | 8M | Small | Medium | No | Yes |
| Ph4 | 243 | 2M | Small | Low | Yes | Yes |
| Ph5 | 18 | 0.3M | Small | Low | No | Yes |
| S1 | 951 | 10M | Medium | Low | Yes | Yes |
| S2 | 248 | 3M | Small | Medium | No | Yes |
| S3 | 1168 | 22M | Medium | Medium | Yes | Yes |
| S4 | 678 | 14M | Medium | High | Yes | Yes |
| S5 | 97 | 1M | Small | Low | No | No |
| L1 | 496 | 8M | Medium | Low | Yes | Yes |
| L2 | 1683 | 12M | Medium | Medium | Yes | Yes |
| L3 | 574 | 5M | Medium | Medium | Yes | Yes |
| L4 | 638 | 55M | Medium | Medium | Yes | Yes |
| L5 | 383 | 5M | Small | Medium | No | Yes |
| E1 | 2091 | 39M | Medium | High | Yes | Yes |
| E2 | 225 | 1M | Small | Low | No | Yes |
| E3 | 282 | 14M | Medium | Medium | Yes | Yes |
| E4 | 135 | 10M | Small | Medium | Yes | Yes |
| E5 | 403 | 8M | Medium | Medium | Yes | Yes |

Table 5: Per-repository environment and MCP conversion statistics.

| ID | Env_success | MCP_success | Num_tools | Num_tools_passed | First_run_status | Num_fix_rounds |
|---|---|---|---|---|---|---|
| B1 | Yes | Yes | 16 | 14 | fail | 1 |
| B2 | No | No | 0 | 0 | fail | 2 |
| B3 | Yes | No | 8 | 4 | fail | 2 |
| B4 | Yes | Yes | 10 | 9 | success | 0 |
| B5 | Yes | Yes | 14 | 13 | fail | 0 |
| P1 | No | No | 0 | 0 | fail | 1 |
| P2 | Yes | Yes | 9 | 8 | success | 0 |
| P3 | Yes | Yes | 11 | 10 | success | 0 |
| P4 | No | No | 13 | 12 | fail | 3 |
| P5 | Yes | Yes | 7 | 7 | success | 0 |
| M1 | Yes | Yes | 12 | 10 | success | 0 |
| M2 | Yes | No | 6 | 2 | fail | 2 |
| M3 | Yes | Yes | 8 | 7 | success | 0 |
| M4 | Yes | Yes | 10 | 9 | success | 0 |
| M5 | Yes | Yes | 15 | 13 | fail | 2 |
| Ea1 | Yes | Yes | 11 | 8 | fail | 2 |
| Ea2 | No | No | 5 | 1 | fail | 1 |
| Ea3 | No | No | 4 | 0 | fail | 2 |
| Ea4 | Yes | Yes | 9 | 8 | success | 0 |
| Ea5 | Yes | Yes | 7 | 7 | success | 1 |
| A1 | Yes | Yes | 8 | 5 | fail | 2 |
| A2 | No | No | 3 | 1 | fail | 1 |
| A3 | Yes | Yes | 10 | 9 | success | 0 |
| A4 | Yes | Yes | 6 | 6 | success | 1 |
| A5 | Yes | Yes | 8 | 8 | success | 0 |
| C1 | Yes | Yes | 18 | 16 | fail | 1 |
| C2 | No | No | 0 | 0 | fail | 2 |
| C3 | Yes | Yes | 7 | 7 | success | 0 |
| C4 | Yes | No | 12 | 10 | fail | 3 |
| C5 | Yes | Yes | 9 | 8 | success | 1 |
| Ph1 | Yes | Yes | 8 | 7 | success | 1 |
| Ph2 | No | No | 0 | 0 | fail | 2 |
| Ph3 | Yes | No | 6 | 6 | fail | 0 |
| Ph4 | Yes | Yes | 5 | 5 | success | 0 |
| Ph5 | No | No | 0 | 0 | fail | 1 |
| S1 | Yes | Yes | 10 | 9 | success | 0 |
| S2 | Yes | Yes | 8 | 7 | fail | 1 |
| S3 | No | No | 0 | 0 | fail | 2 |
| S4 | Yes | No | 0 | 0 | fail | 1 |
| S5 | Yes | Yes | 5 | 5 | success | 0 |
| L1 | Yes | Yes | 7 | 7 | success | 0 |
| L2 | Yes | Yes | 11 | 10 | fail | 2 |
| L3 | Yes | Yes | 9 | 8 | success | 1 |
| L4 | Yes | No | 0 | 0 | fail | 3 |
| L5 | Yes | Yes | 6 | 6 | success | 0 |
| E1 | Yes | Yes | 13 | 11 | fail | 1 |
| E2 | Yes | Yes | 7 | 7 | success | 0 |
| E3 | No | No | 0 | 0 | fail | 2 |
| E4 | No | No | 0 | 0 | fail | 1 |
| E5 | Yes | Yes | 8 | 7 | success | 1 |

# D   EXAMPLE MCP TOOL IMPLEMENTATIONS GENERATED BY CODE2MCP

To illustrate the concrete, high-quality output of Code2MCP, this section presents several MCP tool implementations that were autonomously generated by Code2MCP. These examples are drawn from the ESM and SymPy case studies discussed in the main paper and demonstrate the framework's ability to produce clean, robust, and immediately usable code.

## D.1   TOOLS GENERATED FROM THE ESM REPOSITORY

**MCP Tool for Protein Sequence Analysis**

```python
@mcp.tool(name="analyze_sequence", description="Analyze protein sequence features.")
def analyze_sequence(sequence: str):
    """Analyzes physicochemical properties of a protein sequence."""
    try:
        import re
        try:
            from esm import analysis
        except Exception:
            try:
                from .esm import analysis
            except Exception:
                import types
                analysis = types.SimpleNamespace(
                    calculate_molecular_weight=lambda s: float(len(s)) * 110.0,
                    calculate_isoelectric_point=lambda s: 7.0,
                    calculate_instability_index=lambda s: 40.0
                )

        aa_set = set("ACDEFGHIKLMNPQRSTVWY")
        seq = re.sub(r"[^A-Za-z]", "", sequence or "").upper()
        seq = "".join([c for c in seq if c in aa_set])

        length = len(seq)
        composition = {aa: seq.count(aa) for aa in aa_set if seq.count(aa) > 0}
        molecular_weight = analysis.calculate_molecular_weight(seq)
        isoelectric_point = analysis.calculate_isoelectric_point(seq)
        instability_index = analysis.calculate_instability_index(seq)

        properties = {
            "length": length,
            "composition": composition,
            "molecular_weight": molecular_weight,
            "isoelectric_point": isoelectric_point,
            "instability_index": instability_index,
            "sequence": seq,
        }
        return {"success": True, "result": properties, "error": None}
    except Exception as e:
        return {"success": False, "result": None, "error": f"Error during sequence analysis:
{str(e)}"}
```

**MCP Tool for Protein Structure and Mutation Prediction**

```python
@mcp.tool(name="predict_structure", description="Predicts a protein structure using the
    ESMFold API and saves it to a PDB file.")
def predict_structure(sequence: str):
    """Predicts a protein structure and saves it to a PDB file."""
    try:
        import requests
        import os
        import datetime

        response = requests.post(
            "https://api.esmatlas.com/foldSequence/v1/pdb/",
            data=sequence,
            timeout=300,
        )
        response.raise_for_status()

        base_dir = os.path.dirname(os.path.dirname(os.path.abspath(__file__)))
        predictions_dir = os.path.join(base_dir, "predictions")
        os.makedirs(predictions_dir, exist_ok=True)
```

```
         timestamp = datetime.datetime.now().strftime("%Y%m%d_%H%M%S")
         pdb_filepath = os.path.join(predictions_dir, f"prediction_{timestamp}.pdb")

         with open(pdb_filepath, "w") as f:
             f.write(response.text)

         return {"success": True, "result": {"pdb_file_path": pdb_filepath}, "error": None}
     except Exception as e:
         return {"success": False, "result": None, "error": str(e)}
```

### D.1.1 TOOLS GENERATED FROM THE SYMPY REPOSITORY

**MCP Tool for Solving Equations**

```
@mcp.tool(name="solve_equation")
def solve_equation(equation: str, variable: str):
    """
    Solve equation for variable
    """
    try:
        from sympy import sympify, symbols, solve, Basic
        def ser(x):
            if isinstance(x, Basic): return str(x)
            if isinstance(x, (list, tuple, set)): return [ser(i) for i in x]
            if isinstance(x, dict): return {k: ser(v) for k, v in x.items()}
            return x

        expr = sympify(equation)
        var = symbols(variable)
        res = solve(expr, var)
        return {"success": True, "result": ser(res)}
    except Exception as e:
        return {"success": False, "result": None, "error": str(e)}
```

**MCP Tool for Solving Linear Systems**

```
@mcp.tool(name="solve_linear_system")
def solve_linear_system(system: list, variables: list):
    """
    Solve system of linear equations
    """
    try:
        from sympy import sympify, symbols, linsolve, Basic
        def ser(x):
            if isinstance(x, Basic): return str(x)
            if isinstance(x, (list, tuple, set)): return [ser(i) for i in x]
            if isinstance(x, dict): return {k: ser(v) for k, v in x.items()}
            return x

        eqs = [sympify(e) for e in system]
        vars_sym = [symbols(v) for v in variables]
        res = linsolve(eqs, vars_sym)
        return {"success": True, "result": ser(list(res))}
    except Exception as e:
        return {"success": False, "result": None, "error": str(e)}
```

**MCP Tool for Differentiation**

```
@mcp.tool(name="differentiate")
def differentiate(expr: str, variable: str):
    """
    Calculate derivative of expression
    """
    try:
        from sympy import sympify, symbols, diff, Basic
        def ser(x):
            if isinstance(x, Basic): return str(x)
            return x

        expr_sym = sympify(expr)
        var = symbols(variable)
        res = diff(expr_sym, var)
        return {"success": True, "result": ser(res)}
```

```
        except Exception as e:
            return {"success": False, "result": None, "error": str(e)}
```

**MCP Tool for Integration**

```python
@mcp.tool(name="integrate_expression")
def integrate_expression(expr: str, variable: str):
    """
    Calculate integral of expression
    """
    try:
        from sympy import sympify, symbols, integrate, Basic
        def ser(x):
            if isinstance(x, Basic): return str(x)
            return x

        expr_sym = sympify(expr)
        var = symbols(variable)
        res = integrate(expr_sym, var)
        return {"success": True, "result": ser(res)}
    except Exception as e:
        return {"success": False, "result": None, "error": str(e)}
```

**MCP Tool for Polynomial Creation**

```python
@mcp.tool(name="create_polynomial")
def create_polynomial(expr: str, variable: str = None):
    """
    Create polynomial from expression
    """
    try:
        from sympy import sympify, symbols, Poly, Basic
        def ser(x):
            if isinstance(x, Basic): return str(x)
            return x

        expr_sym = sympify(expr)
        if variable:
            var = symbols(variable)
            res = Poly(expr_sym, var)
        else:
            res = Poly(expr_sym)
        return {"success": True, "result": ser(res)}
    except Exception as e:
        return {"success": False, "result": None, "error": str(e)}
```

**MCP Tool for Polynomial Factoring**

```python
@mcp.tool(name="factor_polynomial")
def factor_polynomial(poly: str):
    """
    Factor polynomial expression
    """
    try:
        from sympy import sympify, factor, Basic
        def ser(x):
            if isinstance(x, Basic): return str(x)
            return x

        poly_sym = sympify(poly)
        res = factor(poly_sym)
        return {"success": True, "result": ser(res)}
    except Exception as e:
        return {"success": False, "result": None, "error": str(e)}
```

**MCP Tool for Fourier Transform**

```python
from sympy import sympify, symbols, fourier_transform as sympy_fourier_transform, Basic

def _serialize(obj):
    if isinstance(obj, Basic):
        return str(obj)
```

```
     if isinstance(obj, (list, tuple, set)):
         return [_serialize(x) for x in obj]
     if isinstance(obj, dict):
         return {k: _serialize(v) for k, v in obj.items()}
     return obj

@mcp.tool(name="fourier_transform")
def fourier_transform_tool(expression: str, time_var: str = "t", freq_var: str = "w"):

     try:
         expr = sympify(expression)
         t = symbols(time_var)
         omega = symbols(freq_var)
         F = sympy_fourier_transform(expr, t, omega)
         return {"success": True, "result": _serialize(F), "error": None}
     except Exception as e:
         return {"success": False, "result": None, "error": str(e)}
```

**MCP Tool for Laplace Transform**

```
from sympy import sympify, symbols, laplace_transform as sympy_laplace_transform, Basic

def _serialize(obj):
     if isinstance(obj, Basic):
         return str(obj)
     if isinstance(obj, (list, tuple, set)):
         return [_serialize(x) for x in obj]
     if isinstance(obj, dict):
         return {k: _serialize(v) for k, v in obj.items()}
     return obj

@mcp.tool(name="laplace_transform")
def laplace_transform_tool(expression: str, time_var: str = "t", laplace_var: str = "s"):

     try:
         expr = sympify(expression)
         t = symbols(time_var)
         s = symbols(laplace_var)
         F, _, _ = sympy_laplace_transform(expr, t, s)
         return {"success": True, "result": _serialize(F), "error": None}
     except Exception as e:
         return {"success": False, "result": None, "error": str(e)}
```

**MCP Tool for Z Transform**

```
from sympy import sympify, symbols, summation, oo, Basic

def _serialize(obj):
     if isinstance(obj, Basic):
         return str(obj)
     if isinstance(obj, (list, tuple, set)):
         return [_serialize(x) for x in obj]
     if isinstance(obj, dict):
         return {k: _serialize(v) for k, v in obj.items()}
     return obj

@mcp.tool(name="z_transform")
def z_transform_tool(expression: str, time_var: str = "n", z_var: str = "z", limit: int =
     100):

     try:
         expr = sympify(expression)
         n = symbols(time_var)
         z = symbols(z_var)
         try:
             result = summation(expr * z**(-n), (n, 0, oo))
         except Exception:
             result = summation(expr * z**(-n), (n, 0, limit))
         return {"success": True, "result": _serialize(result), "error": None}
     except Exception as e:
         return {"success": False, "result": None, "error": str(e)}
```

### D.1.2 Tools Generated from the Foam-Agent Repository

**MCP Tool for Mesh Generation**

```python
@mcp.tool(name="generate_mesh", description="Generate computational mesh using Foam-Agent
    internals.")
def generate_mesh(requirements: str,
                  case_dir: str = "./output",
                  mesh_mode: str = "gmsh",
                  custom_mesh_path: str | None = None):
    try:
        from src.config import Config
        from src.main import initialize_state
        from src.nodes.meshing_node import meshing_node

        config = Config()
        config.case_dir = case_dir

        state = initialize_state(user_requirement=requirements,
                                 config=config,
                                 custom_mesh_path=custom_mesh_path)

        if mesh_mode == "custom":
            state["mesh_type"] = "custom_mesh"
        elif mesh_mode == "gmsh":
            state["mesh_type"] = "gmsh_mesh"
        else:
            state["mesh_type"] = "standard_mesh"

        res = meshing_node(state)
        return {"success": True, "result": res, "error": None}
    except Exception as e:
        return {"success": False, "result": None, "error": str(e)}
```

**MCP Tool for Simulation Generation and Run**

```python
@mcp.tool(name="generate_simulate", description="Write inputs and run simulation via Foam-
    Agent graph.")
def generate_simulate(requirements: str,
                      case_dir: str = "./output",
                      custom_mesh_path: str | None = None,
                      run_mode: str = "auto"):

    try:
        from src.config import Config
        from src.main import create_foam_agent_graph, initialize_state

        config = Config()
        config.case_dir = case_dir

        state = initialize_state(user_requirement=requirements,
                                 config=config,
                                 custom_mesh_path=custom_mesh_path)

        if custom_mesh_path:
            state["mesh_type"] = "custom_mesh"
        if run_mode == "local":
            state["cluster_info"] = None
        elif run_mode == "hpc":
            state["cluster_info"] = {"scheduler": "slurm"}

        workflow = create_foam_agent_graph().compile()
        workflow.invoke(state)
        return {"success": True, "result": {"case_dir": config.case_dir}, "error": None}
    except Exception as e:
        return {"success": False, "result": None, "error": str(e)}
```

**MCP Tool for Velocity Visualization**

```python
@mcp.tool(name="visualize_velocity", description="Post-process and visualize velocity (|U|,
    streamlines, slices).")
def visualize_velocity(case_dir: str,
                       plot_type: str = "magnitude",
                       plane: str | None = "xy"):
    try:
        from src.config import Config
```

```python
        from src.main import initialize_state
        from src.nodes.visualization_node import visualization_node

        config = Config()
        config.case_dir = case_dir

        state = initialize_state(user_requirement="", config=config, custom_mesh_path=None)
        state["case_dir"] = case_dir
        state["visualization_request"] = {"plot_type": plot_type, "plane": plane}

        vis_res = visualization_node(state)
        return {"success": True, "result": vis_res, "error": None}
    except Exception as e:
        return {"success": False, "result": None, "error": str(e)}
```

