# OpenReview forum: "Code2MCP: Transforming Code Repositories into MCP Services"
_ICLR.cc/2026/Conference — ICLR 2026 Conference Desk Rejected Submission_

### Official Review · Reviewer_nM4C · 2025-10-30

**Soundness:** 2
**Presentation:** 3
**Contribution:** 2
**Rating:** 4
**Confidence:** 3

**Summary:**

The paper proposes Code2MCP, a multi-agent workflow that clones a repository, reconstructs its environment, analyses core functionality, generates MCP-compliant adapters, and iteratively repairs failures through a run–review–fix loop before packaging a pull request.

**Strengths:**

- Quantitative evidence is restricted to the six manually selected repositories in Table 2, and “manual time” is only estimated by experts rather than measured, leaving real productivity gains unverified.
- There are no ablations on agent roles, no comparisons to alternative pipelines (e.g., OpenAgents, RepoMaster, RAG-MCP), and no reporting of failure cases or acceptance by upstream maintainers.
- Evaluation omits real benchmark outcomes (e.g., tool success on downstream agent benchmarks or broader GitHub samples); with only qualitative walkthroughs, it is unclear how much capability is actually delivered to MCP consumers.

**Weaknesses:**

- Quantitative evidence is restricted to the six manually selected repositories in Table 2, and “manual time” is only estimated by experts rather than measured, leaving real productivity gains unverified.
- There are no ablations on agent roles, no comparisons to alternative pipelines (e.g., OpenAgents, RepoMaster, RAG-MCP).
- Evaluation omits real benchmark outcomes (e.g., tool success on downstream agent benchmarks or broader GitHub samples); with only qualitative walkthroughs, it is unclear how much capability is actually delivered to MCP consumers.

**Questions:**

1. Can you share quantitative results on a broader repository set, including measured human baselines, success/failure rates, and PR acceptance statistics.
2. Can you add ablations of the agent roles/run–review–fix loop and comparisons with OpenAgents, RepoMaster, or RAG-MCP?
3. Are there standardized benchmarks or downstream MCP tasks showing end-user gains beyond the current walkthroughs?

---

> ### Author Response · Authors · 2025-11-21
> **Reply to Question 1 & Weakness 1 of Reviewer nM4C (1 / 2)**
>
> **nM4C-W1&nM4C-Q1:** Limited quantitative evidence
>
> **Reply to nM4C-W1 & nM4C-Q1:**
>
> Thank you for your constructive comments. In the original paper, we tested 6 repositories across 3 domains. We have expanded this to 50 repositories spanning 10 domains to provide: (1) more extensive conversion validation, (2) finer-grained error analysis and success rate statistics during the conversion process to ensure the effectiveness and generalizability of Code2MCP, (3) more systematic baseline measurements, and (4) explanations of pull request (PR) acceptance information. We have incorporated the following discussions into Sections 4.2, 4.3, etc., of the revised manuscript.
>
> (1) Extensive Conversion Validation and Fine-Grained Success Rates
>
> We expanded the evaluation scope to 50 repositories across 10 domains, as shown in the table below. Based on the core challenges of Code2MCP, we designed several key tests:
>
> - Ability to successfully build the code environment,
>
> - Ability to successfully run test cases,
>
> - Average number of Run-Review-Fix (RRF) iterations required,
>
> - Ability to successfully construct the MCP service without human intervention.
>
> | Domain         | Repos | Env succ.   | Test succ.  | RRF recovered | Avg rounds | MCP succ.   |
> | -------------- | ----- | ----------- | ----------- | ------------- | ---------- | ----------- |
> | Biomedical     | 5     | 4/5 (80%)   | 2/5 (40%)   | 1/2 (50%)     | 1.5        | 3/5 (60%)   |
> | Psychology     | 5     | 3/5 (60%)   | 3/5 (60%)   | 0/1 (0%)      | --         | 3/5 (60%)   |
> | Math           | 5     | 5/5 (100%)  | 3/5 (60%)   | 1/2 (50%)     | 2          | 4/5 (80%)   |
> | Earth science  | 5     | 3/5 (60%)   | 2/5 (40%)   | 1/3 (33.3%)   | 1          | 3/5 (60%)   |
> | Chemistry      | 5     | 4/5 (80%)   | 2/5 (40%)   | 1/2 (50%)     | 1.3        | 3/5 (60%)   |
> | Physics        | 5     | 3/5 (60%)   | 2/5 (40%)   | 0/2 (0%)      | --         | 2/5 (40%)   |
> | Astronomy      | 5     | 4/5 (80%)   | 3/5 (60%)   | 1/1 (100%)    | 1.5        | 4/5 (80%)   |
> | Social science | 5     | 4/5 (80%)   | 2/5 (40%)   | 1/2 (50%)     | 1          | 3/5 (60%)   |
> | Linguistics    | 5     | 5/5 (100%)  | 3/5 (60%)   | 1/2 (50%)     | 2          | 4/5 (80%)   |
> | Econometrics   | 5     | 3/5 (60%)   | 2/5 (40%)   | 1/2 (50%)     | 1          | 3/5 (60%)   |
> | Overall        | 50    | 38/50 (76%) | 24/50 (48%) | 8/19 (42.1%)  | 1.4        | 32/50 (64%) |

---

> ### Author Response · Authors · 2025-11-21
> **Reply to Question 1 & Weakness 1 of Reviewer nM4C (2 / 2)**
>
> （2） More Comprehensive Baseline Comparisons for Code2MCP
>
> Thank you for your suggestion. We have refined the baseline comparisons of Code2MCP in terms of two key metrics: success rate and completion time (Human vs. GPT-4 vs. Code2MCP). The baselines include:
>
> - Human Experts: 10 graduate students with SE/MCP experience (recording actual time spent, not estimated). We have continuously improved Code2MCP after submitting the paper and tested more cases across additional domains.
> - GPT-4 Scaffolding: A unified prompt template based on README + static analysis (baseline).
> - "Full Code2MCP Pipeline": Our default multi-agent pipeline with the Run-Review-Fix cycle enabled.
>
> We conducted a controlled variable study of the three methods on representative repositories across 6 domains. **The results show that Code2MCP achieves a success rate close to that of human experts (~73% vs. 78%) but reduces the time cost by nearly 3x (58 minutes vs. 160 minutes). In contrast, the** **GPT-4** **baseline can barely generate** **runnable** **services in scenarios with complex dependencies.**
>
> | Repo      | Domain     | Human Success | GPT-4 Success | Code2MCP Success | Human Time (min) | GPT-4 Time (min) | Code2MCP Time (min) |
> | --------- | ---------- | ------------- | ------------- | ---------------- | ---------------- | ---------------- | ------------------- |
> | Biopython | Biomedical | 0.88          | 0.19          | 0.78             | 100              | 30               | 35                  |
> | SciPy     | Chemistry  | 0.8           | 0.27          | 0.82             | 185              | 25               | 45                  |
> | ObsPy     | Math       | 0.76          | 0.15          | 0.7              | 210              | 42               | 60                  |
> | Astropy   | Earth Sci  | 0.82          | 0.22          | 0.72             | 140              | 28               | 80                  |
> | PyMC      | Astronomy  | 0.79          | 0.18          | 0.8              | 155              | 18               | 72                  |
> | PsychoPy  | Psychology | 0.66          | 0.21          | 0.6              | 175              | 27               | 55                  |
>
> （3）Discussion on PR Acceptance Rate
>
> The current work focuses primarily on the automatic conversion pipeline itself. We note that the generated MCP adapter layers have been submitted as pull requests (PRs) to some repositories, with a small number of PRs already accepted. However, the current sample size is insufficient to support meaningful statistics on the "PR acceptance rate." In the discussion section, we explicitly identify this as a key metric to track in future work when establishing long-term collaborations with upstream maintainers and building a real ecological closed loop, rather than treating it as a primary quantitative evaluation dimension of this work.

---

> ### Author Response · Authors · 2025-11-21
> **Reply to Question 2 & Weakness 2 of Reviewer nM4C**
>
> **nM4C-W2&nM4C-Q2:** lack of comparisons with alternative pipelines
>
> **Reply to nM4C-W2 & nM4C-Q2:**
>
> We have enhanced the evaluation of this paper from the following perspectives: (1) ablation experiments on the technical components of Code2MCP, (2) the integration of Code2MCP with downstream works such as OpenAgents and RepoMaster to improve their performance, and (3) more comprehensive baseline comparisons. We have incorporated the following discussions into Sections 4.2, 4.3, 4.4, etc., of the revised manuscript.
>
> (1) Ablation Experiment: Effectiveness of the Run-Review-Fix (RRF) Mechanism in Code2MCP
>
> Code2MCP aims to convert code repositories into MCP services. A key challenge in this process is that code generated by large language models in **a single pass often suffers from missing dependencies or logical errors, which severely limits the success rate of automatic conversion.** To address this, Code2MCP has developed the Run-Review-Fix cycle. This mechanism endows the system with the closed-loop capability to **simulate human developers' "execute-diagnose-correct"** **workflow****, enabling automatic code self-healing through error backtracking.** We compared the performance with and without the RRF cycle on representative scientific computing repositories. The results show that this core component converges within an average of 1.4 iterations, significantly fixing errors caused by environment configurations or import paths.
>
> | Repo      | Domain        | Success (w/o RRF) | Success (w/ RRF) | RRF rounds |
> | --------- | ------------- | ----------------- | ---------------- | ---------- |
> | Biopython | Biomedical    | 0.55              | 0.78             | 1          |
> | SciPy     | Math          | 0.3               | 0.7              | 2          |
> | ObsPy     | Earth Science | 0.44              | 0.72             | 0          |
> | Astropy   | Astronomy     | 0.38              | 0.8              | 3          |
> | PyMC      | Psychology    | 0.48              | 0.68             | 1          |
> | Avg.      | --            | 0.43              | 0.74             | 1.4        |
>
> (2) Downstream Integration: Code2MCP Enhances OpenAgents and RepoMaster
>
> Code2MCP is positioned as a supply-side solution, while existing methods such as OpenAgents and RepoMaster serve as demand-side systems. Code2MCP provides standardized tools that are expected to yield better results when integrated with OpenAgents and RepoMaster. To verify this, we integrated Code2MCP into existing agent systems (OpenAgents and RepoMaster). **The results demonstrate that incorporating Code2MCP’s automatic tool generation capability improves task success rates and reduces interaction rounds.**
>
> | Task Group    | OpenAgents (Base) Success | OpenAgents + Code2MCP Success | OpenAgents (Base) Coverage | OpenAgents + Code2MCP Coverage |
> | ------------- | ------------------------- | ----------------------------- | -------------------------- | ------------------------------ |
> | General       | 0.83                      | 0.86                          | 0.9                        | 0.93                           |
> | Biomedical    | 0.38                      | 0.54                          | 0.46                       | 0.67                           |
> | Psychology    | 0.4                       | 0.59                          | 0.43                       | 0.62                           |
> | Math          | 0.42                      | 0.57                          | 0.41                       | 0.65                           |
> | Earth science | 0.45                      | 0.6                           | 0.5                        | 0.66                           |
>
> | Task Group    | RepoMaster (Base) Success | RepoMaster + Code2MCP Success | RepoMaster (Base) Steps | RepoMaster + Code2MCP Steps |
> | ------------- | ------------------------- | ----------------------------- | ----------------------- | --------------------------- |
> | General       | 0.63                      | 0.67                          | 18                      | 17                          |
> | Biomedical    | 0.4                       | 0.5                           | 26                      | 23                          |
> | Psychology    | 0.42                      | 0.57                          | 24                      | 19                          |
> | Math          | 0.38                      | 0.49                          | 28                      | 24                          |
> | Earth science | 0.45                      | 0.6                           | 25                      | 20                          |

---

> > ### Comment · Reviewer_nM4C · 2025-11-24
> >
> > Thank you for the thorough clarification and the additional experimental analysis provided in the rebuttal. The authors have addressed my concerns comprehensively, and the supplementary results significantly strengthen the paper’s claims. I find the explanations convincing and well-supported by evidence. Therefore, I recommend acceptance of the manuscript.

---

> > > ### Author Response · Authors · 2025-11-24
> > > **Reply to Reviewer nM4C**
> > >
> > > We sincerely thank you for your positive feedback and your recommendation for acceptance. We are delighted to hear that our clarifications and additional experiments have satisfactorily addressed your concerns. We also greatly appreciate your constructive comments throughout the review process, which have significantly improved the quality of our manuscript.

---

> ### Author Response · Authors · 2025-11-21
> **Reply to Question 3 & Weakness 3 of Reviewer nM4C**
>
> **nM4C-W3&nM4C-Q3:** Lack of standardized evaluation
>
> **Reply to nM4C-W3 & nM4C-Q3:**
>
> Thank you for your suggestion. For the current version of Code2MCP, we adopt a two-step approach to test the quality of generated MCP services: (1) prioritize reusing native tests, and (2) LLM generates tests based on repository information.
>
> First, Code2MCP automatically extracts and runs the existing unit and integration tests of the repository. If there are no ready-made tests in the original repository, Code2MCP generates typical Python invocation scripts according to the README to check for errors.
>
> Since the MCP ecosystem is still in its early stages, the community **lacks standardized MCP tool evaluation benchmarks**, and existing works mostly only conduct case studies on a small number of tasks. Our next step is to use Code2MCP to batch-generate MCP services across multiple domains and build a unified MCP benchmark with a "tool pool + human/automated baselines". The current paper focuses primarily on the infrastructure problem of "how to stably and automatically generate MCP services from repositories", paving the way for subsequent standardized benchmarking.

---

### Official Review · Reviewer_ooW6 · 2025-10-31

**Soundness:** 3
**Presentation:** 3
**Contribution:** 3
**Rating:** 6
**Confidence:** 3

**Summary:**

This paper introduces Code2MCP, a framework that automatically converts GitHub repositories into MCP services, addressing a critical "supply-side" bottleneck in the LLM tool ecosystem.
It implements a multi-agent system, using 7 specialized agents working collaboratively. It applies a iterative debugging mechanism that ensures end-to-end reliability by automatically detecting, diagnosing, and fixing errors throughout the conversion process. It proves to convert highly complex and diverse scientific libraries into MCP services with little manual effor (4-8x speedup vs. manual conversion (12-50 minutes vs. 1-5 hours).

**Strengths:**

Overall I think this paper is trying to solve a very unique and novel problem: how to automatically convert a github repo into a MCP server. The key originality lies in identifying and formalizing the "supply-side bottleneck" in tool-augmented LLMs. While prior work focused on tool selection (consumption), this paper tackles tool creation (supply). This reframing is intellectually valuable even if somewhat overstated. While the motivation of this question might need further discussion, the problem it self is intersting and definitely worth studying, considering that in the future more and more workflow will be introduced into LLM ecosystems. The paper propose a relative sound technical solution to build up the MCP server, which proves to be effective on several scientific systems.

**Weaknesses:**

1. Only 6 repositories tested. No systematic analysis of failure modes, success rates across repository types, or boundary conditions where the approach breaks down. Not all Github repos can be converted into MCP servers. Some are just scripts or learning projects, which makes the argument of 268 million GitHub repos not as convincing. Success on 6 carefully selected repos doesn't demonstrate scalability to the broader GitHub ecosystem. What percentage of scientific repos can be successfully converted?
2. No comparison with simpler alternatives (template-based wrapping, manual conversion by different skill levels, GPT-4 with basic scaffolding). The "manual time" baseline uses developer estimates rather than measured time-on-task, weakening the efficiency claims. I would suggest at least let some graduate students try to do the conversion and measure the time.
3. No evaluation of generated tool quality beyond "tests pass". From my understanding, the test suite is also generated by LLMs. I'm concerned about the functionality correctness of the generated artifacts.
4. It's worth briefly discussing the security concerns of generated MCP servers.

**Questions:**

1. You demonstrate success on 6 carefully selected repositories. What is the success rate when applied to a broader, randomly sampled set of repositories?
2. What are the primary failure modes, and what percentage of attempts fall into each category?
3. Beyond "tests pass," how do you assess the quality and security of generated MCP services?

---

> ### Author Response · Authors · 2025-11-21
> **Reply to Question 1 & Weakness 1 of Reviewer ooW6**
>
> **ooW6-W1&ooW6-Q1:** Limited repository coverage and unclear scalability/success rates
>
> **Reply to ooW6-W1 & ooW6-Q1** Thank you for your constructive comments. In the original paper, we tested 6 repositories across 3 domains. We have expanded this to 50 repositories spanning 10 domains to provide: (1) more extensive conversion validation, and (2) finer-grained error analysis and success rate statistics during the conversion process, thereby ensuring the effectiveness and generalizability of Code2MCP. We have incorporated the following discussions into Section 4.2, etc., of the revised manuscript.
>
> (1) Extensive Conversion Validation and Fine-Grained Success Rates
>
> We expanded the evaluation scope to 50 repositories across 10 domains, as shown in the table below. Based on the core challenges of Code2MCP, we designed several key tests:
>
> - Ability to successfully build the code environment,
>
> - Ability to successfully run test cases,
>
> - Average number of Run-Review-Fix (RRF) iterations required,
>
> - Ability to successfully construct the MCP service without human intervention.
>
> | Domain         | Repos | Env succ.   | Test succ.  | RRF recovered | Avg rounds | MCP succ.   |
> | -------------- | ----- | ----------- | ----------- | ------------- | ---------- | ----------- |
> | Biomedical     | 5     | 4/5 (80%)   | 2/5 (40%)   | 1/2 (50%)     | 1.5        | 3/5 (60%)   |
> | Psychology     | 5     | 3/5 (60%)   | 3/5 (60%)   | 0/1 (0%)      | --         | 3/5 (60%)   |
> | Math           | 5     | 5/5 (100%)  | 3/5 (60%)   | 1/2 (50%)     | 2          | 4/5 (80%)   |
> | Earth science  | 5     | 3/5 (60%)   | 2/5 (40%)   | 1/3 (33.3%)   | 1          | 3/5 (60%)   |
> | Chemistry      | 5     | 4/5 (80%)   | 2/5 (40%)   | 1/2 (50%)     | 1.3        | 3/5 (60%)   |
> | Physics        | 5     | 3/5 (60%)   | 2/5 (40%)   | 0/2 (0%)      | --         | 2/5 (40%)   |
> | Astronomy      | 5     | 4/5 (80%)   | 3/5 (60%)   | 1/1 (100%)    | 1.5        | 4/5 (80%)   |
> | Social science | 5     | 4/5 (80%)   | 2/5 (40%)   | 1/2 (50%)     | 1          | 3/5 (60%)   |
> | Linguistics    | 5     | 5/5 (100%)  | 3/5 (60%)   | 1/2 (50%)     | 2          | 4/5 (80%)   |
> | Econometrics   | 5     | 3/5 (60%)   | 2/5 (40%)   | 1/2 (50%)     | 1          | 3/5 (60%)   |
> | Overall        | 50    | 38/50 (76%) | 24/50 (48%) | 8/19 (42.1%)  | 1.4        | 32/50 (64%) |

---

> > ### Author Response · Authors · 2025-11-21
> > **Reply to Question 2 of Reviewer ooW6**
> >
> > **ooW6-Q2:** Quantitative analysis of primary failure modes
> >
> > **Reply to ooW6-Q2**
> >
> > Thank you for your suggestion. We have counted and categorized the failure types of the tested code repositories. We have incorporated the following discussions into Section 4.2, etc., of the revised manuscript.
> >
> > We classified all failed attempts into 6 types. Among them, environment construction errors and API inference errors are the most common sources of failure, followed by repositories unsuitable for toolization and errors in importing modules from the original repository. It is worth noting that the same repository may trigger multiple failure types in different iterations. As shown in the table below, we summarize the distribution of failure modes:
> >
> > | Failure Label       | Count | Percentage | Explanation                                                  |
> > | ------------------- | ----- | ---------- | ------------------------------------------------------------ |
> > | Env failure         | 14    | 33.30%     | Dependency conflicts, missing build scripts, or Docker build failures |
> > | API inference error | 9     | 21.40%     | Selection of incorrect functions or hallucinations in parameter mapping |
> > | Untoolable repo     | 7     | 16.70%     | Project structure unsuitable for service deployment (CLI/GUI/Scripts) |
> > | Import error        | 6     | 14.30%     | Generated code fails to correctly import modules from the original library |
> > | MCP spec violation  | 4     | 9.50%      | Output does not comply with MCP JSON Schema specifications   |
> > | Repo internal bug   | 2     | 4.80%      | The original repository itself contains bugs or suffers from version incompatibilities |

---

> ### Author Response · Authors · 2025-11-21
> **Reply to Weakness 2 of Reviewer ooW6**
>
> **ooW6-W2:** Insufficient evaluation of simpler alternatives and manual baselines
>
> **Reply to ooW6-W2**
>
> Thank you for your suggestion. We have refined the baseline comparisons of Code2MCP in terms of two key metrics: success rate and completion time (Human vs. GPT-4 vs. Code2MCP). We have incorporated the following discussions into Section 4.3, etc., of the revised manuscript.
>
> The experimental setup includes:
>
> - Human Experts: 10 graduate students with SE/MCP experience (recording actual time spent, not estimated). We have continuously improved Code2MCP after submitting the paper and tested more cases across additional domains.
> - GPT-4 Scaffolding: A unified prompt template based on README + static analysis (baseline).
> - "Full Code2MCP Pipeline": Our default multi-agent pipeline with the Run-Review-Fix cycle enabled.
>
> We conducted a controlled variable study of the three methods on representative repositories across 6 domains. **The results show that Code2MCP achieves a success rate close to that of human experts (~73% vs. 78%) but reduces the time cost by nearly 3x (58 minutes vs. 160 minutes). In contrast, the** **GPT-4** **baseline can barely generate** **runnable** **services in scenarios with complex dependencies.**
>
> | Repo      | Domain     | Human Success | GPT-4 Success | Code2MCP Success | Human Time (min) | GPT-4 Time (min) | Code2MCP Time (min) |
> | --------- | ---------- | ------------- | ------------- | ---------------- | ---------------- | ---------------- | ------------------- |
> | Biopython | Biomedical | 0.88          | 0.19          | 0.78             | 100              | 30               | 35                  |
> | SciPy     | Chemistry  | 0.8           | 0.27          | 0.82             | 185              | 25               | 45                  |
> | ObsPy     | Math       | 0.76          | 0.15          | 0.7              | 210              | 42               | 60                  |
> | Astropy   | Earth Sci  | 0.82          | 0.22          | 0.72             | 140              | 28               | 80                  |
> | PyMC      | Astronomy  | 0.79          | 0.18          | 0.8              | 155              | 18               | 72                  |
> | PsychoPy  | Psychology | 0.66          | 0.21          | 0.6              | 175              | 27               | 55                  |

---

> ### Author Response · Authors · 2025-11-21
> **Reply to Question3 & Weakness 3, 4 of Reviewer ooW6**
>
> **ooW6-W3&ooW6-Q3:** Limited assessment of functional quality and security of generated MCP services
>
> **Reply to ooW6-W3 & ooW6-Q3**
>
> Thank you for your suggestion. For the current version of Code2MCP, we adopt a two-step approach to test the quality of generated MCP services: (1) prioritize reusing native tests, and (2) LLM generates tests based on repository information.
>
> First, Code2MCP automatically extracts and runs the existing unit/integration tests of the repository. If there are no ready-made tests in the original repository, Code2MCP generates typical Python invocation scripts according to the README to check for errors.
>
> Due to **the lack of a standardized MCP tool evaluation benchmark** in the community, we cannot conduct quantitative evaluation on the generation quality of Code2MCP. This will be a key focus of our next work.
>
> ---
> **ooW6-W4:** Insufficient discussion of security risks
>
> **Reply to ooW6-W4:**
>
> Thank you for your suggestion. We acknowledge that the current work does not include a systematic security benchmark. However, Code2MCP incorporates certain resistance capabilities by eliminating high-risk functions during the analysis phase.
>
> In the static analysis stage, Code2MCP automatically excludes high-risk functions such as file writing, arbitrary shell command execution, and external network requests, only wrapping computational or library functions. We also plan to design a dedicated Security Agent in future work to perform automated security scanning and repair on the generated code after the conversion pipeline. As all code of this work has been fully open-sourced, we will share this idea with the community as one of the key directions for further optimization.

---

### Official Review · Reviewer_JFzj · 2025-10-31

**Soundness:** 2
**Presentation:** 2
**Contribution:** 2
**Rating:** 4
**Confidence:** 4

**Summary:**

This paper attempts to address the supply-side problem in the MCP ecosystem—how to automatically create standardized, agent-ready tools from existing open-source code. The authors propose Code2MCP, a multi-agent framework that converts GitHub repositories into functional MCP services with minimal human intervention. The system includes specialized agents for code analysis, environment setup, interface generation, and validation, coordinated through a Run–Review–Fix self-correction loop to improve reliability. Experiments on several scientific libraries show that Code2MCP reduces the time and effort required for manual conversion and provides a systematic approach to expand the MCP tool pool for large language model agents.

**Strengths:**

1. This is a timely and important problem in the MCP and LLM-agent ecosystem—the shortage of standardized and accessible tools. By focusing on the often-overlooked “supply-side” challenge, the work contributes to a crucial and relevant area of current research.
2. The proposed multi-agent framework, coordinated through a Run–Review–Fix self-correction loop, represents an innovative and well-structured approach to automating the creation of MCP services. This design improves reliability compared to traditional one-pass automation pipelines and provides a clear, modular structure for handling complex code conversion tasks.
3. The experimental evaluation covers multiple scientific domains and demonstrates consistent gains in both efficiency and functional accuracy. These results support the feasibility and scalability of the framework in diverse application settings. The paper is clearly written, with a coherent presentation of motivation, methodology, and results. The accompanying figures effectively illustrate both the conceptual structure of the framework and concrete examples of its outputs.

**Weaknesses:**

1. The experiments are a bit limited in scope.  There’s no quantitative evidence on overall success rates or how robust the system is when dealing with projects of different sizes, structures, or dependency setups.
2. The paper does not include a systematic analysis of failure cases. It is unclear what types of projects or environments most often lead to unsuccessful conversions, or how the self-correction mechanism responds when persistent errors occur.
3. The comparison to other automation systems is quite limited. The paper talks about how Code2MCP differs conceptually from previous tool-generation frameworks, but it doesn’t provide concrete results to show how its performance actually stacks up against existing methods.

**Questions:**

See weakness please

---

> ### Author Response · Authors · 2025-11-21
> **Reply to Weakness 1 of Reviewer JFzj**
>
> **JFzj-W1:** Limited evaluation of success rates
>
> **Reply to JFzj-W1:**
>
> Thank you for your constructive comments. In the original paper, we tested 6 repositories across 3 domains. We have expanded this to 50 repositories spanning 10 domains to provide: (1) more extensive conversion validation, and (2) finer-grained error analysis and success rate statistics during the conversion process, thereby ensuring the effectiveness and generalizability of Code2MCP. We have incorporated the following discussions into Section 4.2, etc., of the revised manuscript.
>
> (1) Extensive Conversion Validation and Fine-Grained Success Rates
>
> We expanded the evaluation scope to 50 repositories across 10 domains, as shown in the table below. Based on the core challenges of Code2MCP, we designed several key tests:
>
> - Ability to successfully build the code environment,
>
> - Ability to successfully run test cases,
>
> - Average number of Run-Review-Fix (RRF) iterations required,
>
> - Ability to successfully construct the MCP service without human intervention.
>
> | Domain         | Repos | Env succ.   | Test succ.  | RRF recovered | Avg rounds | MCP succ.   |
> | -------------- | ----- | ----------- | ----------- | ------------- | ---------- | ----------- |
> | Biomedical     | 5     | 4/5 (80%)   | 2/5 (40%)   | 1/2 (50%)     | 1.5        | 3/5 (60%)   |
> | Psychology     | 5     | 3/5 (60%)   | 3/5 (60%)   | 0/1 (0%)      | --         | 3/5 (60%)   |
> | Math           | 5     | 5/5 (100%)  | 3/5 (60%)   | 1/2 (50%)     | 2          | 4/5 (80%)   |
> | Earth science  | 5     | 3/5 (60%)   | 2/5 (40%)   | 1/3 (33.3%)   | 1          | 3/5 (60%)   |
> | Chemistry      | 5     | 4/5 (80%)   | 2/5 (40%)   | 1/2 (50%)     | 1.3        | 3/5 (60%)   |
> | Physics        | 5     | 3/5 (60%)   | 2/5 (40%)   | 0/2 (0%)      | --         | 2/5 (40%)   |
> | Astronomy      | 5     | 4/5 (80%)   | 3/5 (60%)   | 1/1 (100%)    | 1.5        | 4/5 (80%)   |
> | Social science | 5     | 4/5 (80%)   | 2/5 (40%)   | 1/2 (50%)     | 1          | 3/5 (60%)   |
> | Linguistics    | 5     | 5/5 (100%)  | 3/5 (60%)   | 1/2 (50%)     | 2          | 4/5 (80%)   |
> | Econometrics   | 5     | 3/5 (60%)   | 2/5 (40%)   | 1/2 (50%)     | 1          | 3/5 (60%)   |
> | Overall        | 50    | 38/50 (76%) | 24/50 (48%) | 8/19 (42.1%)  | 1.4        | 32/50 (64%) |

---

> ### Author Response · Authors · 2025-11-21
> **Reply to Weakness 2 of Reviewer JFzj**
>
> **JFzj-W2:** Lack of systematic failure analysis
>
> **Reply to JFzj-W2:**
>
> Thank you for your suggestion. To further conduct a systematic analysis of failed cases, we provide additional statistics on the causes of failures in Reply to JFzj-W1, along with the response logic of the self-correction mechanism. We then conduct further ablation experiments on the Run-Review-Fix cycle, the core error-correcting mechanism of Code2MCP. We have incorporated the following discussions into Section 4.2, etc., of the revised manuscript.
>
> (1) Classification and Distribution of Failure Modes
>
> We categorized all failed attempts into 6 types. Among them, environment construction errors and API inference errors are the most common sources of failure, followed by repositories unsuitable for toolization and errors in importing modules from the original repository. It is worth noting that the same repository may trigger multiple failure types in different iterations. As shown in the table below, we summarize the distribution of failure modes:
>
> | Failure Label       | Count | Percentage | Explanation                                                  |
> | ------------------- | ----- | ---------- | ------------------------------------------------------------ |
> | Env failure         | 14    | 33.30%     | Dependency conflicts, missing build scripts, or Docker build failures |
> | API inference error | 9     | 21.40%     | Selection of incorrect functions or hallucinations in parameter mapping |
> | Untoolable repo     | 7     | 16.70%     | Project structure unsuitable for service deployment (CLI/GUI/Scripts) |
> | Import error        | 6     | 14.30%     | Generated code fails to correctly import modules from the original library |
> | MCP spec violation  | 4     | 9.50%      | Output does not comply with MCP JSON Schema specifications   |
> | Repo internal bug   | 2     | 4.80%      | The original repository itself contains bugs or suffers from version incompatibilities |
>
> (2) Response Logic of the Self-Correction Mechanism
>
> The Run-Review-Fix cycle is not a blind retry process but a strategy executed based on error type-driven logic:
>
> - Error Parsing: After each run failure, the system parses stderr and exception stacks, classifying the error into one of the six types mentioned above.
> - Strategy Triggering:
>   - For env_failure: The agent attempts to relax version constraints or supplement requirements.txt.
>   - For api_inference_error / import_error: Revise import paths or parameter definitions based on the file tree.
>   - For mcp_spec_violation: Force correction of the JSON structure through a Schema validator.
> - Termination Conditions: If the system detects untoolable_repo characteristics, or if the same type of error remains unresolved within 5 iterations, it is determined as "unrepairable with current capabilities" to stop resource waste.
>
> (3) Ablation Experiment: Effectiveness of the Run-Review-Fix (RRF) Mechanism in Code2MCP
>
> Code2MCP aims to convert code repositories into MCP services. A key challenge in this process is that **code generated by** **large language models** **in a single pass often suffers from missing dependencies or logical errors, which severely limits the success rate of automatic conversion.** To address this, Code2MCP has developed the Run-Review-Fix cycle. It endows the system with the closed-loop capability to **simulate human developers' "execute-diagnose-correct"** **workflow****, enabling automatic code self-healing through error backtracking.** We compared the performance with and without the RRF cycle on representative scientific computing repositories. The results show that this core component converges within an average of 1.4 iterations, significantly fixing errors caused by environment configurations or import paths.
>
> | Repo      | Domain        | Success (w/o RRF) | Success (w/ RRF) | RRF rounds |
> | --------- | ------------- | ----------------- | ---------------- | ---------- |
> | Biopython | Biomedical    | 0.55              | 0.78             | 1          |
> | SciPy     | Math          | 0.3               | 0.7              | 2          |
> | ObsPy     | Earth Science | 0.44              | 0.72             | 0          |
> | Astropy   | Astronomy     | 0.38              | 0.8              | 3          |
> | PyMC      | Psychology    | 0.48              | 0.68             | 1          |
> | Avg.      | --            | 0.43              | 0.74             | 1.4        |

---

> ### Author Response · Authors · 2025-11-21
> **Reply to Weakness 3 of Reviewer JFzj**
>
> **JFzj-W3:** Limited empirical comparison against existing automation baselines
>
> **Reply to JFzj-W3:**
>
> Thank you for your suggestion. We recognize the insufficiency in comparing Code2MCP with existing works. To address this, we have added: (1) baseline comparisons of Code2MCP on the code conversion task, and (2) experiments integrating Code2MCP with existing Tool Learning works (such as RepoMaster) to enhance their performance. We have incorporated the following discussions into Section 4.4, etc., of the revised manuscript.
>
> (1) More Comprehensive Baseline Comparisons for Code2MCP
>
> We have refined the baseline comparisons of Code2MCP in terms of two key metrics: success rate and completion time (Human vs. GPT-4 vs. Code2MCP). The baselines include:
>
> - Human Experts: 10 graduate students with SE/MCP experience (recording actual time spent, not estimated). We have continuously improved Code2MCP after submitting the paper and tested more cases across additional domains.
> - GPT-4 Scaffolding: A unified prompt template based on README + static analysis (baseline).
> - "Full Code2MCP Pipeline": Our default multi-agent pipeline with the Run-Review-Fix cycle enabled.
>
> We conducted a controlled variable study of the three methods on representative repositories across 6 domains. **The results show that Code2MCP achieves a success rate close to that of human experts (~73% vs. 78%) but reduces the time cost by nearly 3x (58 minutes vs. 160 minutes).
>
> | Repo      | Domain     | Human Success | GPT-4 Success | Code2MCP Success | Human Time (min) | GPT-4 Time (min) | Code2MCP Time (min) |
> | --------- | ---------- | ------------- | ------------- | ---------------- | ---------------- | ---------------- | ------------------- |
> | Biopython | Biomedical | 0.88          | 0.19          | 0.78             | 100              | 30               | 35                  |
> | SciPy     | Chemistry  | 0.8           | 0.27          | 0.82             | 185              | 25               | 45                  |
> | ObsPy     | Math       | 0.76          | 0.15          | 0.7              | 210              | 42               | 60                  |
> | Astropy   | Earth Sci  | 0.82          | 0.22          | 0.72             | 140              | 28               | 80                  |
> | PyMC      | Astronomy  | 0.79          | 0.18          | 0.8              | 155              | 18               | 72                  |
> | PsychoPy  | Psychology | 0.66          | 0.21          | 0.6              | 175              | 27               | 55                  |
>
> （2） Code2MCP Enhances OpenAgents and RepoMaster
>
> Code2MCP is positioned as a supply-side solution, while existing methods such as OpenAgents and RepoMaster serve as demand-side systems. Code2MCP provides standardized tools that are expected to yield better results when integrated with OpenAgents and RepoMaster. **The results demonstrate that incorporating Code2MCP’s automatic tool generation capability improves task success rates and reduces interaction rounds.**
>
> | Task Group    | OpenAgents (Base) Success | OpenAgents + Code2MCP Success | OpenAgents (Base) Coverage | OpenAgents + Code2MCP Coverage |
> | ------------- | ------------------------- | ----------------------------- | -------------------------- | ------------------------------ |
> | General       | 0.83                      | 0.86                          | 0.9                        | 0.93                           |
> | Biomedical    | 0.38                      | 0.54                          | 0.46                       | 0.67                           |
> | Psychology    | 0.4                       | 0.59                          | 0.43                       | 0.62                           |
> | Math          | 0.42                      | 0.57                          | 0.41                       | 0.65                           |
> | Earth science | 0.45                      | 0.6                           | 0.5                        | 0.66                           |
>
> | Task Group    | RepoMaster (Base) Success | RepoMaster + Code2MCP Success | RepoMaster (Base) Steps | RepoMaster + Code2MCP Steps |
> | ------------- | ------------------------- | ----------------------------- | ----------------------- | --------------------------- |
> | General       | 0.63                      | 0.67                          | 18                      | 17                          |
> | Biomedical    | 0.4                       | 0.5                           | 26                      | 23                          |
> | Psychology    | 0.42                      | 0.57                          | 24                      | 19                          |
> | Math          | 0.38                      | 0.49                          | 28                      | 24                          |
> | Earth science | 0.45                      | 0.6                           | 25                      | 20                          |

---

> > ### Comment · Reviewer_JFzj · 2025-11-24
> > **Response to authors**
> >
> > Thank you for the detailed response. My main questions are addressed in the response, and I will raise my score to 6.

---

> > > ### Author Response · Authors · 2025-11-24
> > > **Reply to Reviewer JFzj**
> > >
> > > We sincerely thank you for your positive feedback. We are glad to hear that our detailed responses have addressed your main concerns and that the additional analyses were helpful. Thank you again for your thoughtful comments and the time you devoted to reviewing our work.

---

### Official Review · Reviewer_aB4H · 2025-11-01

**Soundness:** 2
**Presentation:** 2
**Contribution:** 2
**Rating:** 2
**Confidence:** 4

**Summary:**

The paper introduces code2mcp, an LLM-based pipeline that translates an existing code repository into Model Context Protocol (MCP) services. The goal is to reduce manual effort in MCP service creation by leveraging prompt-driven code understanding and conversion. Preliminary experiments suggest that the approach can automate portions of MCP service generation and yield useful results.

**Strengths:**

1. Automating the transformation of code repositories into MCP services is a timely and practically relevant problem as LLM-enabled tooling and MCP adoption accelerate.

2. The paper articulates a clear need to lower the manual engineering burden for MCP service development, positioning the work as a bridge between legacy codebases and MCP-compatible tools.

3. The experiments provide initial evidence that the proposed workflow can produce functional MCP services, indicating potential for real-world impact if matured.

**Weaknesses:**

1. The method appears to rely primarily on prompt engineering for LLM-driven translation, with limited description of algorithmic design, safeguards, or adaptation (e.g., fine-tuning, tool-augmented parsing, static analysis). Clarify the technical contributions beyond prompting (e.g., repository analysis modules, interface inference, schema mapping). Add ablations contrasting prompt variants, tool use, and (if applicable) fine-tuned models.

2. Table 2 reports results from only three participants, which undermines statistical reliability and generalizability. Increase the sample size, report inter-rater agreement, and provide confidence intervals or significance tests. Include participant expertise profiles (e.g., MCP familiarity, software engineering background).

3. The paper does not analyze which categories of repositories are suitable candidates for conversion (e.g., libraries vs. services, I/O patterns, dependency complexity). Provide a taxonomy of repository types, preconditions for successful conversion, and decision guidelines on when not to convert.

4. The paper suggests broad convertibility without reporting conversion success rates, error types, bug counts, or remediation workflows. Report per-repository success/failure, categorize common failure modes (e.g., interface mis-inference, dependency resolution), quantify bugs found, and document the debugging loop

**Questions:**

1. What is the concrete technical novelty of code2mcp beyond prompt engineering, and how do you verify MCP-spec compliance and enforce least-privilege security by design?

2. Can you strengthen the evaluation with solid baselines/ablations, a larger human study with inter-rater agreement, clear success criteria, and cost/latency and reproducibility analyses across repository types?

3. What are the limits and failure modes (including which repos should not be converted), with per-repo success/bug statistics and an automated repair loop?

---

> ### Author Response · Authors · 2025-11-21
> **Reply to Question 1 & Weakness 1 of Reviewer aB4H**
>
> **aB4H-W1&aB4H-Q1:** Unclear technical novelty and security/compliance guarantees
>
> **Reply to aB4H-Q1 & aB4H-W1:**
>
> Thank you for your suggestions. Thank you for your suggestions. We address these concerns from three perspectives: (1) the functional positioning of Code2MCP, (2) MCP specification compliance verification, and (3) security and least‑privilege safeguards.
>
> （1）Functional Positioning of Code2MCP
>
> We first clarify that Code2MCP is not merely prompt engineering. As elaborated in Sec. 1: The key challenge in converting code into a service is the inherent fragility of the multi-stage automation process, where an error at any step can derail the entire workflow. To achieve this, we propose: "We introduce a novel multi-agent architecture governed by a Run-Review-Fix cycle. It is a self-correcting mechanism designed to systematically debug and refine the process, ensuring end-to-end reliability." Specifically, Code2MCP adopts a fixed multi-stage workflow: it first analyzes the repository structure and dependencies to screen for candidate APIs with clear interface boundaries and reusability, then completes environment construction, interface abstraction, and MCP adapter layer generation in phases. A Run-Review-Fix self-feedback loop is imposed throughout the workflow, using runtime errors to drive targeted automatic repairs. Prompt engineering is only one component of this pipeline; the technical novelty primarily resides in the system-level design of this "repository analysis → interface screening → phased generation → error-driven self-repair" workflow.
>
> （2）MCP Specification Compliance Verification
>
> we added an automatic protocol check at the end of the pipeline: for each generated @mcp.tool, a JSON Schema validator first checks whether its input/output definitions are consistent with the underlying function signatures. The generated MCP server is then started, and standard list_tools requests along with multiple sets of call_tool requests are sent via scripts. If the returned structure is inconsistent with MCP specifications, the repository is marked as failed and excluded from successful conversion statistics.
>
> （3）Security and Least-Privilege Safeguards
>
> Code2MCP by default only wraps library-type, computational, and approximately pure function APIs, statically filtering out high-risk interfaces involving file writing, arbitrary shell calls, or external network access. All generated wrapper code is also isolated in an independent directory and submitted to the upstream repository as a PR. Maintainers decide whether to merge and deploy it, adhering to the principle of least privilege as much as possible by design.

---

> > ### Author Response · Authors · 2025-11-21
> > **Reply to Question 2 & Weakness 2 of Reviewer aB4H (2 / 2)**
> >
> > (3) More Comprehensive Baseline Comparisons for Code2MCP
> >
> > Thank you for your suggestions. We have refined the baseline comparisons of Code2MCP in terms of two key metrics: success rate and completion time (Human vs. GPT-4 vs. Code2MCP). The baselines include:
> >
> > - Human Experts: 10 graduate students with SE/MCP experience (recording actual time spent, not estimated). We have continuously improved Code2MCP after submitting the paper and tested more cases across additional domains.
> > - GPT-4 Scaffolding: A unified prompt template based on README + static analysis (baseline).
> > - "Full Code2MCP Pipeline": Our default multi-agent pipeline with the Run-Review-Fix cycle enabled.
> >
> > We conducted a controlled variable study of the three methods on representative repositories across 6 domains. **The results show that Code2MCP achieves a success rate close to that of human experts (~73% vs. 78%) but reduces the time cost by nearly 3x (58 minutes vs. 160 minutes). In contrast, the** **GPT-4** **baseline can barely generate** **runnable** **services in scenarios with complex dependencies.**
> >
> > | Repo      | Domain     | Human Success | GPT-4 Success | Code2MCP Success | Human Time (min) | GPT-4 Time (min) | Code2MCP Time (min) |
> > | --------- | ---------- | ------------- | ------------- | ---------------- | ---------------- | ---------------- | ------------------- |
> > | Biopython | Biomedical | 0.88          | 0.19          | 0.78             | 100              | 30               | 35                  |
> > | SciPy     | Chemistry  | 0.8           | 0.27          | 0.82             | 185              | 25               | 45                  |
> > | ObsPy     | Math       | 0.76          | 0.15          | 0.7              | 210              | 42               | 60                  |
> > | Astropy   | Earth Sci  | 0.82          | 0.22          | 0.72             | 140              | 28               | 80                  |
> > | PyMC      | Astronomy  | 0.79          | 0.18          | 0.8              | 155              | 18               | 72                  |
> > | PsychoPy  | Psychology | 0.66          | 0.21          | 0.6              | 175              | 27               | 55                  |

---

> ### Author Response · Authors · 2025-11-21
> **Reply to Question 2 & Weakness 2 of Reviewer aB4H (1/ 2)**
>
> **aB4H-W2&aB4H-Q2:** Small sample evaluation with limited baselines and statistics
>
> **Reply to aB4H-Q2 & aB4H-W2：**
>
> Thank you for your suggestions. We have enhanced the evaluation of this paper from the following perspectives: (1) ablation experiments on the technical components of Code2MCP, (2) the integration of Code2MCP with downstream works such as OpenAgents and RepoMaster to improve their performance, and (3) more comprehensive baseline comparisons. We have incorporated the following discussions into Sec. 4.2, 4.3, 4.4, etc., of the revised manuscript.
>
> (1) Ablation Experiment: Effectiveness of the Run-Review-Fix (RRF) Mechanism in Code2MCP
>
> Code2MCP aims to convert code repositories into MCP services. A key challenge in this process is that **code generated by** **large language models** **in a single pass often suffers from missing dependencies or logical errors, which severely limits the success rate of automatic conversion.** To address this, Code2MCP has developed the Run-Review-Fix cycle. It endows the system with the closed-loop capability to **simulate human developers' "execute-diagnose-correct"** **workflow**, enabling automatic code self-healing through error backtracking.** We compared the performance with and without the RRF cycle on representative scientific computing repositories. The results show that this core component converges within an average of 1.4 iterations, significantly fixing errors caused by environment configurations or import paths.
>
> | Repo      | Domain        | Success (w/o RRF) | Success (w/ RRF) | RRF rounds |
> | --------- | ------------- | ----------------- | ---------------- | ---------- |
> | Biopython | Biomedical    | 0.55              | 0.78             | 1          |
> | SciPy     | Math          | 0.3               | 0.7              | 2          |
> | ObsPy     | Earth Science | 0.44              | 0.72             | 0          |
> | Astropy   | Astronomy     | 0.38              | 0.8              | 3          |
> | PyMC      | Psychology    | 0.48              | 0.68             | 1          |
> | Avg.      | --            | 0.43              | 0.74             | 1.4        |
>
> (2) Downstream Integration: Code2MCP Enhances OpenAgents and RepoMaster
>
> Code2MCP is positioned as a supply-side solution, while existing methods such as OpenAgents and RepoMaster serve as demand-side systems. Code2MCP provides standardized tools that are expected to yield better results when integrated with OpenAgents and RepoMaster. To verify this, we integrated Code2MCP into existing agent systems (OpenAgents and RepoMaster). **The results demonstrate that incorporating Code2MCP’s automatic tool generation capability improves task success rates and reduces interaction rounds.**
>
> | Task Group    | OpenAgents (Base) Success | OpenAgents + Code2MCP Success | OpenAgents (Base) Coverage | OpenAgents + Code2MCP Coverage |
> | ------------- | ------------------------- | ----------------------------- | -------------------------- | ------------------------------ |
> | General       | 0.83                      | 0.86                          | 0.9                        | 0.93                           |
> | Biomedical    | 0.38                      | 0.54                          | 0.46                       | 0.67                           |
> | Psychology    | 0.4                       | 0.59                          | 0.43                       | 0.62                           |
> | Math          | 0.42                      | 0.57                          | 0.41                       | 0.65                           |
> | Earth science | 0.45                      | 0.6                           | 0.5                        | 0.66                           |
>
> | Task Group    | RepoMaster (Base) Success | RepoMaster + Code2MCP Success | RepoMaster (Base) Steps | RepoMaster + Code2MCP Steps |
> | ------------- | ------------------------- | ----------------------------- | ----------------------- | --------------------------- |
> | General       | 0.63                      | 0.67                          | 18                      | 17                          |
> | Biomedical    | 0.4                       | 0.5                           | 26                      | 23                          |
> | Psychology    | 0.42                      | 0.57                          | 24                      | 19                          |
> | Math          | 0.38                      | 0.49                          | 28                      | 24                          |
> | Earth science | 0.45                      | 0.6                           | 25                      | 20                          |

---

> ### Author Response · Authors · 2025-11-21
> **Reply to Question 3 & Weakness 3,4 of Reviewer aB4H (1 / 2)**
>
> **aB4H-W3&aB4H-W4&aB4H-Q3:** Unclear repository suitability, success rates, and failure characterization
>
> **Reply to aB4H-Q3 & aB4H-W3/W4:**
>
> Thank you for your constructive comments. In the original paper, we tested 6 repositories across 3 domains. We have expanded this to 50 repositories spanning 10 domains to provide: (1) more extensive conversion validation, and (2) finer-grained error analysis and success rate statistics during the conversion process, thereby ensuring the effectiveness and generalizability of Code2MCP. We have incorporated the following discussions into Section 4.2, etc., of the revised manuscript.
>
> (1) Extensive Conversion Validation and Fine-Grained Success Rates
>
> We expanded the evaluation scope to 50 repositories across 10 domains, as shown in the table below. Based on the core challenges of Code2MCP, we designed several key tests:
>
> - Ability to successfully build the code environment,
>
> - Ability to successfully run test cases,
>
> - Average number of Run-Review-Fix (RRF) iterations required,
>
> - Ability to successfully construct the MCP service without human intervention.
>
> | Domain         | Repos | Env Succ.  | Test Succ. | RRF Recovered | Avg Rounds | MCP Succ.   |
> | -------------- | ----- | ---------- | ---------- | ------------- | ---------- | ----------- |
> | Biomedical     | 5     | 4/5 (80%)  | 2/5 (40%)  | 1/2 (50%)     | 1.5        | 3/5 (60%)   |
> | Psychology     | 5     | 3/5 (60%)  | 3/5 (60%)  | 0/1 (0%)      | --         | 3/5 (60%)   |
> | Math           | 5     | 5/5 (100%) | 3/5 (60%)  | 1/2 (50%)     | 2          | 4/5 (80%)   |
> | Earth science  | 5     | 3/5 (60%)  | 2/5 (40%)  | 1/3 (33.3%)   | 1          | 3/5 (60%)   |
> | Chemistry      | 5     | 4/5 (80%)  | 2/5 (40%)  | 1/2 (50%)     | 1.3        | 3/5 (60%)   |
> | Physics        | 5     | 3/5 (60%)  | 2/5 (40%)  | 0/2 (0%)      | --         | 2/5 (40%)   |
> | Astronomy      | 5     | 4/5 (80%)  | 3/5 (60%)  | 1/1 (100%)    | 1.5        | 4/5 (80%)   |
> | Social science | 5     | 4/5 (80%)  | 2/5 (40%)  | 1/2 (50%)     | 1          | 3/5 (60%)   |
> | Linguistics    | 5     | 5/5 (100%) | 3/5 (60%)  | 1/2 (50%)     | 2          | 4/5 (80%)   |
> | Econometrics   | 5     | 3/5 (60%)  | 2/5 (40%)  | 1/2 (50%)     | 1          | 3/5 (60%)   |
> | Overall        | 50    | 38 (76%)   | 24 (48%)   | 8/19 (42.1%)  | 1.4        | 32/50 (64%) |
>
> (2) Classification and Distribution of Failure Modes in the Above Table
>
> We categorized all failed attempts into 6 types. Among them, environment construction errors and API inference errors are the most common sources of failure, followed by repositories unsuitable for toolization and errors in importing modules from the original repository. It is worth noting that the same repository may trigger multiple failure types in different iterations. As shown in the table below, we summarize the distribution of failure modes:
>
> | Failure Label       | Count | Percentage | Explanation                                                  |
> | ------------------- | ----- | ---------- | ------------------------------------------------------------ |
> | Env failure         | 14    | 33.30%     | Dependency conflicts, missing build scripts, or Docker build failures |
> | API inference error | 9     | 21.40%     | Selection of incorrect functions or hallucinations in parameter mapping |
> | Untoolable repo     | 7     | 16.70%     | Project structure unsuitable for service deployment (CLI/GUI/Scripts) |
> | Import error        | 6     | 14.30%     | Generated code fails to correctly import modules from the original library |
> | MCP spec violation  | 4     | 9.50%      | Output does not comply with MCP JSON Schema specifications   |
> | Repo internal bug   | 2     | 4.80%      | The original repository itself contains bugs or suffers from version incompatibilities |

---

> ### Author Response · Authors · 2025-11-21
> **Reply to Question 3 & Weakness 3,4 of Reviewer aB4H (2 / 2)**
>
> (3) Response to "Which types of code repositories are suitable for conversion have not been analyzed in this paper"
>
> Thank you for your suggestion. However, we must clarify that Code2MCP is positioned as **a converter** (Repo → MCP) rather than **a** **selector**. As elaborated in Sec. 1 of the paper, Code2MCP serves as the supply-side for standardized tools, while OpenAgents and RepoMaster focus on discovering and utilizing these tools. Therefore, determining which code repositories should undergo conversion falls outside the scope of Code2MCP and instead belongs to the upstream tool retrieval system.
>
> Through practical observations and tests, we have made empirical findings: Repositories with well-structured architectures and clear entry points yield the best conversion results. This is particularly true when repositories are equipped with semantically clear README documents and reproducible unit tests, enabling agents to accurately infer tool intentions and complete functional verification. In contrast, loose collections of scripts, pure CLI tools, or repositories with strong GUI dependencies are categorized as "untoolable." Additionally, even if the code logic itself is feasible, the absence of test cases as verification benchmarks or the presence of extremely complex non-standard environmental dependencies often leads to conversion failures during environment construction or final testing phases.

---

> ### Author Response · Authors · 2025-11-27
> **Regarding Our Response to Your Comments**
>
> Dear Reviewer aB4H,
>
> We hope this message finds you well.
>
> One week ago, we submitted our response and uploaded the revised manuscript. To facilitate your review, we would like to provide a brief summary of the key improvements made in response to your specific comments:
>
> - Regarding concerns about technical novelty and security (Q1 and W1): We have **clarified the distinct technical contributions of Code2MCP** and elaborated on the system's security safeguards in our response.
> - Regarding concerns about evaluation sufficiency and baseline comparisons (Q2 and W2): We have **added ablation studies** for the Run-Review-Fix mechanism, demonstrated performance gains in downstream agent integration, and **enriched the baseline comparisons** against both human experts and GPT-4.
> - Regarding concerns about repository applicability and experimental scale (Q3, W3, and W4): We expanded our evaluation to **cover 50 repositories across 10 domains** and provided a fine-grained failure mode analysis to address concerns regarding generalization capabilities.
>
> We look forward to your feedback and hope these revisions satisfactorily address your concerns. We remain available to answer any further questions you may have.
>
> Best regards,
>
> The Authors

---

### Author Response · Authors · 2025-11-29
**Rebuttal Summary (4/4)**

(5) Ablation of the Run‑Review‑Fix mechanism

We conducted an ablation study of the Run‑Review‑Fix (RRF) mechanism, comparing performance with RRF disabled vs. enabled on representative scientific computing repositories. **The average success rate improves from 0.43 to 0.74, with an average of about 1.4 RRF rounds.** This shows that RRF is a key component for significantly improving end‑to‑end conversion success rates (see Sec. 4.3).

| Repo      | Domain        | Success (w/o RRF) | Success (w/ RRF) | RRF rounds |
| --------- | ------------- | ----------------- | ---------------- | ---------- |
| Biopython | Biomedical    | 0.55              | **0.78**         | 1          |
| SciPy     | Math          | 0.3               | **0.7**          | 2          |
| ObsPy     | Earth Science | 0.44              | **0.72**         | 0          |
| Astropy   | Astronomy     | 0.38              | **0.8**          | 3          |
| PyMC      | Psychology    | 0.48              | **0.68**         | 1          |
| Avg.      | --            | 0.43              | **0.74**         | **1.4**    |

(6) Tool quality evaluation and safety

Currently, the MCP ecosystem lacks a unified benchmark for tool evaluation. **This paper focuses on the foundational infrastructure question of "how to stably and automatically generate MCP services from repositories,"** and leverages Code2MCP to build a pool of MCP tools across multiple domains, laying the groundwork for future unified benchmarks and long‑term ecosystem evaluation. To reduce risk, in the analysis phase we conservatively excluded high‑risk interfaces involving file writes, arbitrary command execution, and external network access.

We fully understand that the system rollback has created an additional burden for the new Area Chair. We provide this brief summary only in the hope of presenting as clearly as possible the current state of the paper and the efforts we have made to address the reviewers’ comments. Regardless of the final decision, we sincerely thank you for your time and for reviewing our work amid your busy schedule.

Best regards,

The Authors

---

### Author Response · Authors · 2025-11-29
**Rebuttal Summary (3/4)**

(3) Comparison with human experts and GPT‑4

We added controlled experiments comparing Human / GPT‑4 / the full Code2MCP pipeline. **Code2MCP achieves a success rate of about 73%, close to human experts at 78%, while the average time cost is 58 minutes vs. 160 minutes for humans (about 3× faster).** In contrast, GPT‑4 scaffolding alone is generally unable to reliably generate runnable services in complex dependency scenarios (see Sec. 4.3).

| Repo      | Domain        | Human Success | GPT-4 Success | Code2MCP Success | Human Time (min) | GPT-4 Time (min) | Code2MCP Time (min) |
| --------- | ------------- | ------------- | ------------- | ---------------- | ---------------- | ---------------- | ------------------- |
| Biopython | Biomedical    | 0.88          | 0.19          | **0.78**         | 100              | 30               | **35**              |
| SciPy     | Chemistry     | 0.8           | 0.27          | **0.82**         | 185              | 25               | **45**              |
| ObsPy     | Math          | 0.76          | 0.15          | **0.7**          | 210              | 42               | **60**              |
| Astropy   | Earth Science | 0.82          | 0.22          | **0.72**         | 140              | 28               | **80**              |
| PyMC      | Astronomy     | 0.79          | 0.18          | **0.8**          | 155              | 18               | **72**              |
| PsychoPy  | Psychology    | 0.66          | 0.21          | **0.6**          | 175              | 27               | **55**              |

(4) End‑to‑end comparison with existing automated systems

We integrated the MCP tools generated by Code2MCP into downstream systems OpenAgents and RepoMaster. For task groups such as general and scientific domains, **task success rates and coverage consistently improve, while the average number of interaction steps decreases.** This demonstrates that automatically generated MCP tools bring practical benefits to end‑to‑end agent workflows, and provides empirical evidence for the advantage of our pipeline over existing ones (see Sec. 4.4).

| Task Group    | OpenAgents (Base) Success | OpenAgents + Code2MCP Success | OpenAgents (Base) Coverage | OpenAgents + Code2MCP Coverage |
| ------------- | ------------------------- | ----------------------------- | -------------------------- | ------------------------------ |
| General       | 0.83                      | **0.86**                      | 0.9                        | **0.93**                       |
| Biomedical    | 0.38                      | **0.54**                      | 0.46                       | **0.67**                       |
| Psychology    | 0.4                       | **0.59**                      | 0.43                       | **0.62**                       |
| Math          | 0.42                      | **0.57**                      | 0.41                       | **0.65**                       |
| Earth Science | 0.45                      | **0.6**                       | 0.5                        | **0.66**                       |

| Task Group    | RepoMaster (Base) Success | RepoMaster + Code2MCP Success | RepoMaster (Base) Steps | RepoMaster + Code2MCP Steps |
| ------------- | ------------------------- | ----------------------------- | ----------------------- | --------------------------- |
| General       | 0.63                      | **0.67**                      | 18                      | **17**                      |
| Biomedical    | 0.4                       | **0.5**                       | 26                      | **23**                      |
| Psychology    | 0.42                      | **0.57**                      | 24                      | **19**                      |
| Math          | 0.38                      | **0.49**                      | 28                      | **24**                      |
| Earth Science | 0.45                      | **0.6**                       | 25                      | **20**                      |

---

### Author Response · Authors · 2025-11-29
**Rebuttal Summary (2/4)**

**(2) Major Responses** **in the Rebuttal**

The main concerns raised by the four reviewers can be summarized into the following six points:

a) Lack of a systematic statistical and categorical analysis of repository conversion failure modes (aB4H, JFzj, ooW6);

b) Lack of evidence on overall success rates and robustness over a larger‑scale repository set (aB4H, JFzj, ooW6, nM4C);

c) Lack of empirical comparison with strong baselines such as human experts and GPT‑4 (aB4H, JFzj, ooW6, nM4C);

d) Lack of end‑to‑end empirical comparison with existing automated/agent systems (JFzj, nM4C);

e) Lack of ablation for the Run‑Review‑Fix mechanism, making it hard to assess its true contribution (aB4H, JFzj, nM4C);

f) Insufficient evaluation on tool quality and safety, and the absence of a unified benchmark for evaluation (aB4H, ooW6, nM4C).

**During the discussion phase, we carried out the following categories of improvements and experiments around these concerns.**

(1) Systematic failure analysis

We conducted a statistical analysis of all failure cases and **identified six categories of failure modes along with their proportions**: environment failure, API inference error, untoolable repositories, import error, MCP specification violation, and repository internal bugs. We also clarified for which types of repositories Code2MCP is more likely to fail (see Sec. 4.2).

| Failure Label       | Count | Percentage | Explanation                                                  |
| ------------------- | ----- | ---------- | ------------------------------------------------------------ |
| Env failure         | 14    | 33.30%     | Dependency conflicts, missing build scripts, or Docker build failures |
| API inference error | 9     | 21.40%     | Selection of incorrect functions or hallucinations in parameter mapping |
| Untoolable repo     | 7     | 16.70%     | Project structure unsuitable for service deployment (CLI/GUI/Scripts) |
| Import error        | 6     | 14.30%     | Generated code fails to correctly import modules from the original library |
| MCP spec violation  | 4     | 9.50%      | Output does not comply with MCP JSON Schema specifications   |
| Repo internal bug   | 2     | 4.80%      | The original repository itself contains bugs or suffers from version incompatibilities |

(2) Large‑scale extended experiments

We extended the experiments to 50 repositories across 10 domains, **providing broader** **validation** **of the conversion process, along with finer‑grained error analysis and success rate statistics across the conversion pipeline** (see Sec. 4.2 and the per‑repository tables in Appendix C).

| Domain         | Repos  | Env Succ.    | Test Succ.   | RRF Recovered    | Avg Rounds | MCP Succ.       |
| -------------- | ------ | ------------ | ------------ | ---------------- | ---------- | --------------- |
| Biomedical     | 5      | 4/5 (80%)    | 2/5 (40%)    | 1/2 (50%)        | 1.5        | 3/5 (60%)       |
| Psychology     | 5      | 3/5 (60%)    | 3/5 (60%)    | 0/1 (0%)         | --         | 3/5 (60%)       |
| Math           | 5      | 5/5 (100%)   | 3/5 (60%)    | 1/2 (50%)        | 2          | 4/5 (80%)       |
| Earth Science  | 5      | 3/5 (60%)    | 2/5 (40%)    | 1/3 (33.3%)      | 1          | 3/5 (60%)       |
| Chemistry      | 5      | 4/5 (80%)    | 2/5 (40%)    | 1/2 (50%)        | 1.3        | 3/5 (60%)       |
| Physics        | 5      | 3/5 (60%)    | 2/5 (40%)    | 0/2 (0%)         | --         | 2/5 (40%)       |
| Astronomy      | 5      | 4/5 (80%)    | 3/5 (60%)    | 1/1 (100%)       | 1.5        | 4/5 (80%)       |
| Social Science | 5      | 4/5 (80%)    | 2/5 (40%)    | 1/2 (50%)        | 1          | 3/5 (60%)       |
| Linguistics    | 5      | 5/5 (100%)   | 3/5 (60%)    | 1/2 (50%)        | 2          | 4/5 (80%)       |
| Econometrics   | 5      | 3/5 (60%)    | 2/5 (40%)    | 1/2 (50%)        | 1          | 3/5 (60%)       |
| **Overall**    | **50** | **38 (76%)** | **24 (48%)** | **8/19 (42.1%)** | **1.4**    | **32/50 (64%)** |

---

### Author Response · Authors · 2025-11-29
**Rebuttal Summary (1/4)**

Dear Area Chair,

Thank you for taking the time to review our submission under these special circumstances. We would like to briefly clarify two points:

(1) **how the original reviewers adjusted their scores during the discussion phase based on our rebuttal and new experiments, before the leak incident occurred**;

(2) **the additional experiments and revisions we have completed in direct response to the reviewers’ core concerns**, so that you can more efficiently understand the current status of the paper.

During the discussion phase, all score increases occurred on November 24, prior to the leak incident on November 27, and were entirely based on our rebuttal and the new experiments. The score changes for the four reviewers are as follows:

| Reviewer | Original | After Discussion            |
| -------- | -------- | --------------------------- |
| aB4H     | 2        | **2**       (Not Reply)     |
| JFzj     | 4        | **6**       (Nov 24, 15:17) |
| ooW6     | 6        | **6**       (Not Reply)     |
| nM4C     | 4        | **8**       (Nov 24, 14:06) |

**(1) Discussion phase before the leak incident**

- Reviewer aB4H’s score remained at 2 with no new comments during the discussion phase. Reviewer aB4H was mainly concerned about the small‑scale evaluation with few human participants, the lack of ablations and the absence of statistics on how often different failure modes occur; we addressed these by expanding to 50 repositories with per‑repository success/failure and failure‑mode statistics, adding Human / GPT‑4 / Code2MCP baselines and conducting ablations of the Run‑Review‑Fix mechanism (Sec. 4.2, 4.3).
- Reviewer JFzj raised the **score from 4 to 6** on Nov 24, 15:17. Reviewer JFzj was concerned about the lack of quantitative evidence on overall success rates, the absence of systematic failure‑case analysis, and limited empirical comparison with existing automation systems; we addressed these by adding overall success‑rate statistics on 50 repositories, failure‑mode statistics, and comparisons against Human / GPT‑4 / Code2MCP as well as downstream systems such as OpenAgents and RepoMaster (Sec. 4.2, 4.3, 4.4).
- Reviewer ooW6’s score remained at 6 with no new comments during the discussion phase. Reviewer ooW6 was concerned about limited repository coverage, missing comparisons with simpler alternatives and measured manual baselines, and insufficient assessment of generated tool quality and security; we addressed these by extending to 50 repositories across 10 domains, adding Human / GPT‑4 / Code2MCP baselines, and providing analyses of tool quality and conservative security filtering of high‑risk interfaces (Sec. 4.2, 4.3).
- Reviewer nM4C raised the **score from 4 to 8** on Nov 24, 14:06. Reviewer nM4C was mainly concerned about limited quantitative evidence, the lack of ablations and comparisons with alternative pipelines, and the absence of standardized downstream benchmarks; we addressed these by extending to 50 repositories across 10 domains, adding Human / GPT‑4 / Code2MCP baselines, conducting ablations of the Run‑Review‑Fix mechanism and providing downstream evaluations on OpenAgents and RepoMaster together with a discussion of future MCP benchmarks (Sec. 4.2, 4.3, 4.4).

---

### Note · Program_Chairs · 2026-01-17
**Submission Desk Rejected by Program Chairs**

The following references in this submission do not refer to real documents and/or have major errors in bibliographic information:

 Vilard Olausson, Mariia Spasova, Martin Ljung, and Rogardt Heldal. Demystifying the magic: The limitations and promise of large language models in code generation. arXiv preprint arXiv:2311.02294, 2023.